# NonZero: Interaction-Guided Exploration for Multi-Agent Monte Carlo Tree Search

**Sizhe Tang** [1]   **Zuyuan Zhang** [1]   **Mahdi Imani** [2]   **Tian Lan** [1]

## Abstract

Monte Carlo Tree Search (MCTS) scales poorly in cooperative multi-agent domains because expansion must consider an exponentially large set of joint actions, severely limiting exploration under realistic search budgets. We propose NONZERO, which keeps multi-agent MCTS tractable by running surrogate-guided selection over a low-dimensional nonlinear representation using an interaction-guided proposal rule, instead of directly exploring the full joint-action space. Our exploration uses an interaction score: single-agent deviations are ranked by predicted gain, while two-agent deviations are scored by a mixed-difference measure that reveals coordination benefits even when no single agent can improve alone. We formalize candidate proposal as a bandit problem over local deviations and derive a proposal rule, NONUCT, with a sublinear local-regret guarantee for reaching approximate graph-local optima without enumerating the joint-action space. Empirically, NONZERO improves sample efficiency and final performance on MatGame, SMAC, and SMACv2 relative to strong model-based and model-free baselines under matched search budgets.

## 1. Introduction

Monte Carlo Tree Search (MCTS) is a widely used planning method, with successes in game playing (Sironi et al., 2018; Silver et al., 2017; Schrittwieser et al., 2020), robotics (Leisiazar et al., 2023), and combinatorial optimization (Xiao et al., 2023). Its effectiveness in standard MCTS stems from Upper Confidence Bound for Trees (UCT) selection, which concentrates computation on promising parts of the search tree while still exploring uncertain actions through confidence bonuses (Kocsis & Szepesvári, 2006). When combined with learned predictors, as in AlphaZero (Silver et al., 2017) and MuZero (Schrittwieser et al., 2020), MCTS can be applied effectively in large state spaces by refining decisions online under a fixed simulation budget.

Scaling MCTS to *cooperative multi-agent* planning is challenging because action selection is combinatorial. With $n$ agents and $d$ actions per agent, the joint-action set has size $|\mathcal{A}| = d^n$, so naive expansion induces an exponential branching factor and quickly exhausts practical simulation budgets (Lowe et al., 2017; Hernandez-Leal et al., 2019; Kwak et al., 2024; Liu et al., 2024). This bottleneck is especially challenging when returns exhibit strong interaction effects, where high-value outcomes may require coordinated deviations that are unlikely to be found by uninformed sampling.

Prior work alleviates this burden only partially by addressing different aspects of the problem. MAZero (Liu et al., 2024) improves model learning and distributed planning components, but tree expansion still fundamentally depends on which joint actions are selected at each node. MALinZero (Tang et al., 2025) reduces joint-action search by exploiting linear return structure, enabling efficient exploration when improvements arise from independent single-agent changes; however, this assumption can miss coordinated improvements when returns are non-additive. Related value factorization approaches, such as VDN (Sunehag et al., 2017) and QMIX (Rashid et al., 2020b), similarly impose structural constraints on the joint value, but they are not designed to support the uncertainty-aware action expansion required by tree search.

We propose NONZERO, a multi-agent MCTS framework that keeps search tractable without enumerating the $|\mathcal{A}| = d^n$ joint-action space. At each node, NONZERO runs surrogate-guided selection over a low-dimensional nonlinear representation using an interaction-guided proposal rule. To drive candidate expansion, NONZERO fits a compact low-dimensional nonlinear predictor of joint-action returns and evaluates structured one- and two-agent deviations from the current candidates. Single-agent deviations are prioritized by predicted gain, while coordinated two-agent deviations are scored using a mixed-difference interaction measure

---

[1]The George Washington University [2]Northeastern University. Correspondence to: Sizhe Tang <s.tang1@gwu.edu>.

*Proceedings of the 43rd International Conference on Machine Learning*, Seoul, South Korea. PMLR 306, 2026. Copyright 2026 by the author(s).

that targets coordination benefits even when no single agent can improve alone (coordination traps), enabling targeted coordination search while keeping the per-node branching factor fixed.

We formalize this candidate-expansion step as a nonlinear bandit problem over local deviations and derive NONUCT, an optimistic proposal rule for allocating deviation queries during exploration. Under discrete smoothness assumptions on the return surrogate, we obtain a sublinear local-regret guarantee for reaching approximate graph-local optima with complexity that does not scale with $|\mathcal{A}| = d^n$. Empirically, integrating NONZERO into MuZero-style planning yields consistent gains in both sample efficiency and final performance on MatGame, SMAC (Samvelyan et al., 2019), and SMACv2 (Ellis et al., 2023) relative to strong model-based and model-free baselines under matched search budgets.

Our main contributions are: (i) NONZERO, a candidate-set multi-agent MCTS framework that overcomes the combinatorial joint-action bottleneck through interaction-guided candidate expansion based on a compact nonlinear return surrogate. (ii) NONUCT, a principled proposal mechanism derived from a nonlinear bandit formulation over local deviations, enabling coordinated exploration without enumerating the $d^n$ joint-action space. (iii) A sublinear local-regret analysis under graph-local optimality, together with consistent empirical gains on MatGame, SMAC, and SMACv2 under matched search budgets.

## 2. Related Works

We formalize the cooperative multi-agent planning task as a Decentralized Partially Observable Markov Decision Process (Dec-POMDP) (Oliehoek et al., 2016), represented by the tuple $\mathcal{M} = \langle \mathcal{I}, \mathcal{S}, \mathcal{A}, P, R, \Omega, \mathcal{O}, \gamma \rangle$. Here, $\mathcal{I} = \{1, \ldots, n\}$ denotes the set of $n$ agents operating within a global state space $\mathcal{S}$. At each discrete time step $t$, every agent $i \in \mathcal{I}$ receives a local observation $o_t^i \in \Omega_i$ and subsequently selects a local action $a_t^i$ from its individual action set $\mathcal{A}_i$ based on its action-observation history. These individual choices constitute a *joint action* vector $\mathbf{a}_t = (a_t^1, \ldots, a_t^n)$, which resides in the joint action space $\mathcal{A} \equiv \prod_{i=1}^n \mathcal{A}_i$. Upon the execution of $\mathbf{a}_t$, the environment transitions to a new state $s_{t+1} \sim P(\cdot|s_t, \mathbf{a}_t)$ and yields a shared global reward $r_t = R(s_t, \mathbf{a}_t)$. The collective objective of the agents is to identify a joint policy $\boldsymbol{\pi}$ that maximizes the expected cumulative discounted return, formally defined as $J(\boldsymbol{\pi}) = \mathbb{E}_{\boldsymbol{\pi}}[\sum_{t=0}^{\infty} \gamma^t r_t]$, where $\gamma \in [0, 1)$ serves as the discount factor.

**MARL with factorized representations.** Factorization-based methods are pivotal for mitigating the curse of dimensionality in the joint action space. Under the Centralized Training with Decentralized Execution (CTDE)

paradigm, VDN (Sunehag et al., 2017) approximates the global value function $Q_{\text{tot}}$ via a linear sum of local utilities. This approach has been extended to monotonic non-linear combinations in QMIX (Rashid et al., 2020b) and nearly-decomposable structures in NDQ (Wang et al., 2019). Further advancements include policy-based factorizations (DOP (Wang et al., 2020), FOP (Zhang et al., 2021)) and value-correction mechanisms (QTRAN (Son et al., 2019), WQMIX (Rashid et al., 2020a)) to address representational limitations. However, their structural assumptions (e.g., additivity or monotonicity) are insufficient to capture complex interactions inherent in sophisticated multi-agent tasks.

**MCTS-based Planning.** MCTS has been established as a powerful paradigm for sequential decision-making in complex planning horizons (Browne et al., 2012; Silver et al., 2017). Instead of relying on exhaustive search, MCTS approximates optimal values through iterative simulation, typically involving four stages: *Selection*, *Expansion*, *Simulation*, and *Back-Propagation*. To eliminate the dependency on ground-truth simulators, model-based variants like MuZero (Schrittwieser et al., 2020) introduce learnable components: a representation model $h_\theta$, a dynamics model $g_\theta$, and a prediction model $f_\theta$. Given an observation history, MuZero maps it to a latent state via $h_\theta$, and performs recursive predictions using $g_\theta$ and $f_\theta$. Crucially, during the *Selection* phase, MuZero employs the Probabilistic Upper Confidence Tree (pUCT) rule to balance exploration and exploitation:

$$a_t = \arg\max_{a \in \mathcal{A}} \left[ \Phi(s, a) + c(s) P(s, a) \frac{\sqrt{\sum_b N(s, b)}}{1 + N(s, a)} \right], \tag{1}$$

where $\Phi(s, a)$ is the estimated Q-value, $P(s, a)$ is the prior policy probability, and $c(s)$ is a scaling factor. This makes evaluating and maintaining selection statistics over Eq. 1 over the full joint-action space computationally intractable.

To address large action spaces, Sampled MuZero (Hubert et al., 2021) modifies the algorithm to only search over a randomly sampled subset of actions $T(s) \subset \mathcal{A}$, drawn from a proposal policy $\beta$. The selection rule is adjusted to $a_t = \arg\max_{a \in T(s)} \left[ \Phi(s, a) + c(s) \frac{\hat{\beta}(a)}{\beta(a)} P(s, a) \frac{\sqrt{\sum_{b \in T(s)} N(s, b)}}{1 + N(s, a)} \right]$ with an importance sampling correction. Although this sampling strategy (and similar distributed approaches like MAZero (Liu et al., 2024)) makes execution feasible, it fundamentally relies on the quality of the proposal distribution $\beta$. In the vast combinatorial landscapes of multi-agent tasks, random or heuristic sampling is often inefficient, failing to cover sparse optimal joint actions. In contrast, NONZERO constructs proposals using a learned nonlinear return surrogate and interaction-guided local deviations, targeting coordinated joint actions without relying on uninformed sampling.

# 3. NONZERO for Multi-Agents MCTS

NONZERO leverages low-dimensional representations of the joint-action returns and solves the resulting nonlinear bandit problem to enable efficient and effective NONUCT-based MCTS in complex multi-agent planning. NONUCT is applied as a proposal and learning mechanism that guides candidate expansion and local surrogate refinement during MCTS. NONZERO follows MuZero with (i) representation, (ii) dynamics, and (iii) prediction heads, and adds (iv) a mixing/hypernetwork that initializes the node-wise surrogate parameter $\theta$ used by NONUCT. To achieve more accurate and efficient reward approximation, we propose NONUCT to capture the non-linearity of low-dimensional rewards and analyze its regret. All proofs are collected in the Appendix A. Here, "low-dimensional" refers to the fact that exploration and estimation depend on local deviations in $\mathcal{A}$ rather than on enumerating the full joint-action set $|\mathcal{A}| = d^n$.

## 3.1. Leveraging Low-Dimensional Nonlinear Representations

NONZERO models the joint-action returns through a low-dimensional nonlinear combination of the latent per-agent action rewards, based on which an online learner optimizes to capture the representation. Precisely, we consider a deterministic nonlinear bandit problem with an unknown parameter $\theta^* \in \Theta$ and a discrete joint-action set $\mathcal{A} \subset \{0,1\}^{nd}$ consisting of $n$-hot vectors (one-hot per agent), where $n$ denotes the number of agents and $d$ is the size of action set for each agent. Thus, each joint action $a \in \mathcal{A}$ is represented by an $n$-hot vector selecting one local action for each agent. Given the action $a$, the environment reveals a deterministic reward $r_t = \eta(\theta^*, a)$.

We define first- and second-order *finite-difference* operators on the joint-action adjacency graph. Let $a = (a_1, \ldots, a_n) \in \mathcal{A}$ and define the one-agent neighbor $a^{(i \leftarrow j)}$ as the joint action obtained by changing only agent $i$'s action to $j \in \mathcal{A}_i$. Let the neighbor set be $\mathcal{N}(a) = \{a^{(i \leftarrow j)} : i \in [n], j \in \mathcal{A}_i, j \neq a_i\}$. We denote a discrete direction by $u = (i \leftarrow j)$ and write $a^{(u)} := a^{(i \leftarrow j)}$. The first-order directional difference is

$$\Delta_u \eta(\theta, a) := \eta(\theta, a^{(u)}) - \eta(\theta, a).$$

For two directions $u = (i \leftarrow j)$ and $v = (k \leftarrow \ell)$ with $i \neq k$, define the mixed second-order difference

$$\Delta^2_{u,v} \eta(\theta, a) := \eta(\theta, a^{(u,v)}) - \eta(\theta, a^{(u)}) - \eta(\theta, a^{(v)}) + \eta(\theta, a),$$

where $a^{(u,v)}$ applies both one-agent changes. The identity $\eta(\theta, a^{(u,v)}) - \eta(\theta, a) = \Delta_u \eta(\theta, a) + \Delta_v \eta(\theta, a) + \Delta^2_{u,v} \eta(\theta, a)$ makes $\Delta^2_{u,v}$ a discrete interaction/curvature signal for coordinated deviations.

In this work, we define a score function $z(\theta, a) = \langle w(\theta), \psi(a) \rangle$ where $w(\theta) \in \mathbb{R}^{nd}$ and $\psi(a) \in \mathbb{R}^{nd}$. We

adopt the asinh link $g(z) = c \operatorname{asinh}(\alpha z)$ with fixed scale parameters $c > 0$ and $\alpha > 0$, and define the estimated deterministic reward $\eta(\theta, a) = g(z(\theta, a))$. Thus, for each $n$-hot action vector $a \in \mathcal{A}$, we model the low-dimensional reward $\eta(\theta, a)$ as a nonlinear combination of $n$ corresponding per-agent action rewards. This procedure can be viewed as projecting the actual reward $r_t$ into the low-dimensional space representable with $\eta(\theta^*, a) = c \operatorname{asinh}(\alpha \langle w(\theta^*), \psi(a) \rangle)$. It reduces MCTS from considering $d^n$ joint reward values in each state to learning a structured parameter vector whose estimation complexity is independent of the exponential joint-action count $|\mathcal{A}| = d^n$, thereby allowing quick estimation of the global reward structure from limited samples and speeding up tree search convergence in multi-agent MCTS. Using the learned return proxy $\eta(\hat{\theta}_t, a)$, we select an action

$$a_t = \arg \max_{a \in \mathcal{A}} \eta(\hat{\theta}_t, a). \tag{2}$$

Directly solving (2) over the full joint-action set $\mathcal{A}$ is intractable when $|\mathcal{A}| = d^n$. Instead, at each tree node $s$ we maintain a small candidate set $\mathcal{C}(s) \subset \mathcal{A}$ of expanded joint actions. MCTS selection compares only actions in $\mathcal{C}(s)$ using the learned surrogate score $\eta(\hat{\theta}_s, \cdot)$. NONUCT is used only as a proposal mechanism that adds a small number of new candidates to $\mathcal{C}(s)$ by applying one-agent or two-agent deviations predicted to yield the largest discrete improvement under $\eta(\hat{\theta}_s, \cdot)$. This keeps the per-node branching factor $|\mathcal{C}(s)|$ fixed while still allowing coordinated exploration.

However, the global convergence of the nonlinear bandit problem mentioned above is statistically intractable. Specifically, it requires exponential samples in the input dimension; for example, $\mathcal{O}(|\mathcal{A}|) = \mathcal{O}(d^n)$ globally optimizing a nonlinear reward model over $\mathcal{A}$ requires samples that scaling with the joint-action count $|\mathcal{A}| = d^n$. Thus, we propose guiding the model to converge to a local maximum rather than directly seeking global optimization. Note that the loss function $\tilde{\eta}(\theta, a)$ is not concave, but its discrete local-optimality structure admits efficient convergence guarantees under our smoothness assumptions (Assumption 3.2), without requiring global maximization over $\mathcal{A}$. We do not assume global concavity; our guarantees are graph-local and hold under one-agent and two-agent deviations.

In general, globally optimizing a nonlinear reward model over the combinatorial set $\mathcal{A}$ is intractable (Dong et al., 2021). Accordingly, our analysis targets efficient convergence to *graph-local* optima, measured by gains in one-agent and two-agent deviations (Definition 3.1). This is the relevant notion for scalable multi-agent planning, where coordinated improvements may require deviations across agents. To achieve this, we need to augment the loss function so that our model satisfies $\eta(\hat{\theta}_t, a_t) \approx \eta(\theta^*, a_t)$, $\Delta_u \eta(\hat{\theta}, a_t) \approx \Delta_u \eta(\theta^*, a_t)$, and $\Delta^2_{u,v} \eta(\hat{\theta}, a_t) \approx$

$\Delta_{u,v}^2 \eta(\theta^*, a_t)$. It is implied that the virtual reward $\eta(\hat{\theta}_t, a_t)$ will keep improving until $a_t$ is a local maxima of $\eta(\theta^*, a_t)$ if we optimize the learning process via taking greedy actions.

Since the real environment provides only the realized reward for the executed joint action, we compute all counterfactual evaluations needed for discrete differences from the learned reward model at the node state (MuZero-style planning model):

$$\Delta_u \eta(\theta^*, a) := \eta(\theta^*, a^{(u)}) - \eta(\theta^*, a). \tag{3}$$

For discrete curvature, we compute the mixed second-order difference:

$$
\begin{aligned}
\Delta_{u,v}^2 \eta(\theta^*, a) := {} & \eta(\theta^*, a^{(u,v)}) - \eta(\theta^*, a^{(u)}) \\
& - \eta(\theta^*, a^{(v)}) + \eta(\theta^*, a).
\end{aligned}
\tag{4}
$$

where $v$ is a random sampled direction independent of $u$. Note that the number of projections above does not scale with the exponential branching factor $d^n$; it depends only on the statistical complexity of the learned surrogate class and the number of sampled deviation directions per update, since only the norm of the projections needs to be estimated. This action-dimension-free property ensures that the proposed NONUCT can achieve efficient exploration in the $d^n$-large action space.

$\Delta_u \eta(\theta, a)$ measures the gain from a single-agent deviation, while $\Delta_{u,v}^2 \eta(\theta, a)$ measures the non-additive interaction (synergy/anti-synergy) of two coordinated deviations. In cooperative MARL, local one-agent improvements may be negative even when a coordinated two-agent deviation is beneficial; $\Delta_{u,v}^2$ provides a principled signal for proposing such coordinated joint actions without enumerating $\mathcal{A}$. Given $x = (a, u, v)$, define the prediction target

$$\hat{y}(\theta, x) = \left[ \eta(\theta, a), \eta(\theta, a^{(u)}), \Delta_u \eta(\theta, a), \Delta_{u,v}^2 \eta(\theta, a) \right] \tag{5}$$

where $a'$ denotes the action perturbation used to produce the first and second order differences. The supervision is

$$y(x) = \left[ \eta(\theta^*, a), \eta(\theta^*, a^{(u)}), \Delta_u \eta(\theta^*, a), \Delta_{u,v}^2 \eta(\theta^*, a) \right] \tag{6}$$

And we note that $\hat{y}(\theta, x), y(x) \in \mathbb{R}^4$.

In MCTS, we treat the learned reward head $r_\psi(s, \cdot)$ at a fixed node state $s$ as the *planning-time* reward function. All counterfactual quantities (e.g., $r_\psi(s, a^{(u)})$ and $r_\psi(s, a^{(u,v)})$) are obtained by evaluating the learned model, without additional environment interaction. Accordingly, $\theta^*$ denotes the best-in-class parameter (within our asinh-GLM family) that locally fits $r_\psi(s, \cdot)$ around the candidate actions explored at node $s$. Only the selected legal joint action is executed in the real environment for data collection.

The loss function that needs to be optimized is defined by

$$
\begin{aligned}
\mathcal{L}_{\text{NONUCT}} := {} & \min_\theta \ \mathbb{E}_{a,u,v} \Big[ \frac{1}{4} \big( (\eta(\theta, a) - \eta(\theta^*, a))^2 \\
& + (\eta(\theta, a^{(u)}) - \eta(\theta^*, a^{(u)}))^2 + (\Delta_u \eta(\theta, a) - \Delta_u \eta(\theta^*, a))^2 \\
& + (\Delta_{u,v}^2 \eta(\theta, a) - \Delta_{u,v}^2 \eta(\theta^*, a))^2 \big) \Big].
\end{aligned}
\tag{7}
$$

### 3.2. Analyzing Regret

In this section, we derive the theoretical guarantees of NONUCT. Unlike standard UCT, which relies on global optimism (often intractable in high-dimensional spaces, i.e., requiring $\mathcal{O}(e^{nd})$ samples), NONUCT exploits the curvature information of the asinh-activated reward landscape. Our analysis targets convergence to graph-local maximizers under one-agent and two-agent deviations, which is the appropriate notion of optimality for scalable multi-agent planning.

We quantify the performance via cumulative local regret, defined as the cumulative gap between the reward of the selected action $a_t$ and the reward of the nearest approximate local maximizer. We prove that NONZERO achieves a sub-linear regret $\hat{R}_T = \tilde{O}(T^{3/4})$, ensuring efficient convergence. The regret is measured with respect to the first time an $(\varepsilon_1, \varepsilon_2)$-local maximizer is reached, counting only steps taken outside this set.

**Geometric and Smoothness Assumptions.** We first define the convergence target and the regularity conditions of the reward landscape.

**Definition 3.1** (Approximate local maximizer on the joint-action graph)**.** Let $\tilde{\eta}(\theta^*, a)$ be the objective over the discrete set $\mathcal{A}$. Define the one-agent deviation gain $G_1(a) := \max_u \Delta_u \tilde{\eta}(\theta^*, a)$ and the best two-agent coordinated deviation gain $G_2(a) := \max_{u \neq v} \big( \tilde{\eta}(\theta^*, a^{(u,v)}) - \tilde{\eta}(\theta^*, a) \big)$. We say $a$ is an $(\varepsilon_1, \varepsilon_2)$-approximate local maximizer if $G_1(a) \leq \varepsilon_1$ and $G_2(a) \leq \varepsilon_2$.

The role of the first and second order differences in Eq. 7 is to refine the node-specific parameter $\hat{\theta}_s$ so that the optimistic score $\eta(\hat{\theta}_s, a)$ used in Eq. 2 is locally accurate around candidate actions, improving exploration efficiency without enumerating the full joint-action space.

While finding global optima is generally NP-hard in non-convex settings, the invexity properties of the asinh-GLM landscape imply that approximate local maxima are effectively global or optimism-equivalent (Kalai & Sastry, 2009). Thus, convergence to the state defined above serves as a robust proxy for optimal planning. The choice of the asinh link function is pivotal to our theoretical framework. Unlike sigmoidal activations that suffer from vanishing gradients (saturation) or ReLU networks that lack higher-order

smoothness, the asinh function, $g(z) = c \cdot \mathrm{asinh}(\alpha z)$, is strictly increasing, unbounded, and infinitely differentiable. Crucially, its derivatives decay at a controlled polynomial rate, ensuring that the reward landscape is sufficiently smooth to allow curvature-based optimization. This structural property justifies the following smoothness assumptions, which are satisfied by the asinh-GLM class by design.

**Assumption 3.2** (Discrete smoothness on $\mathcal{A}$)**.** There exist constants $\zeta_1, \zeta_2, \zeta_3 > 0$ such that for all $\theta \in \Theta$, all $a \in \mathcal{A}$, and all feasible directions $u, v, w$ (with distinct agents when needed),

$$|\Delta_u \eta(\theta, a)| \le \zeta_1, \ |\Delta_{u,v}^2 \eta(\theta, a)| \le \zeta_2,$$
$$|\Delta_{u,v}^2 \eta(\theta, a^{(w)}) - \Delta_{u,v}^2 \eta(\theta, a)| \le \zeta_3. \tag{8}$$

Assumption 3.2 formalizes the boundedness of discrete gains and interaction effects under single- and two-agent deviations, which holds naturally for bounded reward models.

We establish the regret bound by decomposing the regret into two parts: a geometric improvement term, which quantifies the guaranteed increase in reward towards an approximate local maximizer, and a statistical estimation term, which captures the prediction discrepancy controlled by the sequential Rademacher complexity. We denote the composite prediction error at step $t$ as a squared error over the four scalar supervision targets (value at $a$, value at a one-agent neighbor, one-agent directional difference, and two-agent mixed difference):

$$\begin{aligned} \xi_t := & \left( \eta(\theta_t, a_t) - \eta(\theta^\star, a_t) \right)^2 \\ & + \left( \eta(\theta_t, a_t^{(u_t)}) - \eta(\theta^\star, a_t^{(u_t)}) \right)^2 \\ & + \left( \Delta_{u_t} \eta(\theta_t, a_t) - \Delta_{u_t} \eta(\theta^\star, a_t) \right)^2 \\ & + \left( \Delta_{u_t,v_t}^2 \eta(\theta_t, a_t) - \Delta_{u_t,v_t}^2 \eta(\theta^\star, a_t) \right)^2. \end{aligned} \tag{9}$$

which encompasses the discrepancies in reward values and in first-/second-order discrete differences used for local improvement.

**Lemma 3.3** (One-step Geometric Improvement)**.** *Let $f(a) = \tilde{\eta}(\theta^*, a)$ and let $\xi_t$ be the prediction error in Eq. 9. Provided that the current action is not yet an approximate local maximizer (i.e., $a_t \notin \mathcal{A}_{\epsilon, 6\sqrt{\zeta_{3rd}\epsilon}}$), the discrete curvature-guided proposal yields, in expectation over the proposal sampling, a guaranteed reward ascent:*

$$\mathbb{E}[f(a_{t+1}) - f(a_t)] \ge \nu(\epsilon) - C_1 \cdot \xi_{t+1}, \tag{10}$$

*where $\nu(\epsilon) > 0$ is the strictly positive improvement rate and $C_1 > 0$ is a constant.*

Lemma 3.3 establishes a sufficient ascent property: the algorithm ensures a strict increase in the ground-truth reward at each step, modulo the online learner's estimation error. We now provide bounds on the estimation error.

**Lemma 3.4** (Estimation Error Control)**.** *The cumulative prediction error is bounded by the generalization capacity of the learner:*

$$\sum_{t=1}^{T} \xi_t \le \sqrt{T \cdot \mathcal{R}_T}. \tag{11}$$

*In* NONZERO*, we have $\mathcal{R}_T = \tilde{O}(\sqrt{T})$, which implies the cumulative error scales sub-linearly as $\tilde{O}(T^{3/4})$.*

Now we verify the convergence. Summing the one-step improvement over $T$ iterations relates the total reward gain to the cumulative estimation error:

$$\underbrace{\tilde{\eta}(\theta^*, a_T) - \tilde{\eta}(\theta^*, a_0)}_{\text{Total Gain}} = \sum_{t=0}^{T-1} (\tilde{\eta}(\theta^*, a_{t+1}) - \tilde{\eta}(\theta^*, a_t))$$
$$\ge \sum_{t=0}^{T-1} \nu(\epsilon) \cdot \mathbb{I}(a_t \notin \mathcal{A}_\epsilon) - C_1 \sum_{t=1}^{T} \xi_t. \tag{12}$$

Since the reward is bounded, the total gain is finite. This inequality implies that the algorithm cannot stay outside the local maximizer set $\mathcal{A}_\epsilon$ for too long, unless the estimation error is large. Formalizing this intuition leads to the following regret bound.

**Theorem 3.5** (Regret Bound of NONUCT)**.** *For any tolerance $\epsilon \le \min(1, \zeta_{3rd}/16)$, the expected cumulative regret of* NONUCT *with respect to finding $(\epsilon, 6\sqrt{\zeta_{3rd}\epsilon})$-local maxima is bounded by:*

$$\mathbb{E}[Regret_T] \le (1 + C_1\sqrt{4TR_T}) \cdot \mathcal{K}(\epsilon), \tag{13}$$

*where $\mathcal{K}(\epsilon) = 1/\nu(\epsilon)$ encapsulates the complexity regarding the landscape difficulty and target precision.*

**Corollary 3.6** (The Order of Regret Bound for NONUCT)**.** *Under the above conditions, the cumulative regret bound of* NONUCT *satisfies:*

$$\mathbb{E}[Regret_T] = \tilde{O}\left(T^{3/4}\right). \tag{14}$$

The regret bound of NONUCT in Theorem 3.5 is action-dimension free while sublinear. We quantify the efficiency gap as follows to demonstrate NONZERO's exploration efficiency.

**Theorem 3.7** (Efficiency Separation)**.** *Let $T_{NonUCT}(d, \epsilon)$ and $T_{UCB}(d, \epsilon)$ denote the number of rounds required to output an approximate local maximizer. The separation ratio $\zeta_{sep}$ satisfies:*

$$\zeta_{sep} \triangleq \frac{T_{UCB}(d, \epsilon)}{T_{NonUCT}(d, \epsilon)} \ge \frac{\exp(c \cdot nd)}{poly(nd, \epsilon^{-1})}, \tag{15}$$

*where $c > 0$ is a universal constant.*

NONUCT achieves action-dimension-free convergence in the sense that its sample complexity scales polynomially with the model's statistical capacity, whereas standard UCB suffers from an unavoidable curse of dimensionality in the joint action space.

### 3.3. Algorithm Framework

---

**Algorithm 1** NONZERO MCTS with Discrete Curvature-Guided Proposals

---

1: **Input:** root state $s_0$
2: **Hyperparameters:** simulations $N_{\text{sim}}$, candidate budget $K$, learning rate $\gamma$
3: **for** sim $= 1$ to $N_{\text{sim}}$ **do**
4: $\quad path \leftarrow \text{SELECT}(s_0)$
5: $\quad leaf \leftarrow \text{last}(path)$
6: $\quad$ **if** $leaf$ not terminal **then**
7: $\quad\quad \text{EXPAND}(leaf)$
8: $\quad$ **end if**
9: $\quad \text{BACKUP}(path)$
10: **end for**
11: **return** $\pi(s_0)$ from visit counts

---

12: **Procedure** SELECT($node$)
13: **while** $node$ has children **do**
14: $\quad a^\star \leftarrow \arg\max_{a \in \mathcal{C}(node)} \quad \eta(\theta_{node}, a)$
15: $\quad node \leftarrow node.\text{child}(a^\star)$
16: $\quad$ append $node$ to $path$
17: **end while**
18: **return** $path$

---

19: **Procedure** EXPAND($node$)
20: Sample $\mathcal{C}(node)$ with policy-prior samples
21: **for all** $a \in \mathcal{C}(node)$ **do**
22: $\quad node' \leftarrow node.\text{child}(a)$
23: $\quad \theta \leftarrow \text{HYPERNETWORK}(node'.state)$
24: $\quad$ Sample $u = (i \leftarrow j), v = (k \leftarrow \ell), i \neq k$
25: $\quad$ Form $a^{(u)}, a^{(u,v)}$
26: $\quad \widehat{\Delta}_u \leftarrow \eta(\theta, a^{(u)}) - \eta(\theta, a)$
27: $\quad \widehat{\Delta}^2_{u,v} \leftarrow \eta(\theta, a^{(u,v)}) - \eta(\theta, a)$
28: **end for**

---

29: **Procedure** BACKUP($path$)
30: **for** $node$ in $path$ (reversed) **do**
31: $\quad$ Update $N(node, \cdot)$ and $Q(node, \cdot)$
32: $\quad$ Sample $u = (i \leftarrow j), v = (k \leftarrow \ell), i \neq k$
33: $\quad$ Pick $a \in \mathcal{C}(node)$
34: $\quad$ Compute $r(a), r(a^{(u)}), r(a^{(u,v)})$
35: $\quad \Delta_u r \leftarrow r(a^{(u)}) - r(a)$
36: $\quad \Delta^2_{u,v} r \leftarrow r(a^{(u,v)}) - r(a^{(u)}) - r(a^{(v)}) + r(a)$
37: $\quad \theta_{node} \leftarrow \theta_{node} - \gamma \nabla_\theta L_{\text{NONUCT}}$
38: **end for**

---

Integrated with curvature-aware learning, NONZERO achieves efficient exploitation and exploration in complex MCTS. The NONZERO model is unrolled with four key

phases as standard MCTS and trained in an end-to-end manner like MuZero (Schrittwieser et al., 2020). The detailed pseudocode of NonZero is shown in Algorithm 1.

NONZERO augments the standard MCTS lifecycle by replacing the UCB heuristic in selection with a curvature-aware optimization routine. In the *Selection* and *Back-Propagation* phase, rather than relying on scalar confidence bounds that scale poorly in combinatorial spaces, the algorithm selects among the currently expanded child actions using the learned surrogate optimized by NONUCT. This step exploits the asinh-induced smoothness to efficiently navigate the joint-action space, guiding the search path toward the $(\epsilon_g, \epsilon_h)$-approximate local maximizers defined in Section 3.2. This mechanism allows the agent to synthesize complex multi-agent coordination strategies without exhaustively enumerating the exponential action space.

To ensure the reward landscape $\eta(\hat{\theta}, \cdot)$ accurately reflects the ground truth, NONZERO incorporates a two-stage estimation strategy. Subsequently, during *Back-propagation*, the algorithm performs online adaptation to refine this estimate. By collecting the reward evaluations and estimating the local geometry via the random projection operators (Eq. 3 and 4), we update the node-specific parameter $\hat{\theta}$ by minimizing the curvature loss $\mathcal{L}_{\text{NONUCT}}$ via gradient descent. This ensures that the local approximation progressively aligns with the true environmental dynamics as the simulation depth increases.

**Hypernetworks for parameter estimation warm-up.** To achieve faster convergence of $\theta$, the NONZERO model involves a hypernetwork (Ha et al., 2016) to predict the initialization of the state-wise $\theta$ when a new node is added to the search tree. Therefore, the gathered interaction information to estimate $\theta$ will be accumulated throughout the training process, which accelerates the convergence of NONZERO. Specifically, the hypernetwork takes the state $s_{t+k}$ as the input and generates the parameter $\theta$. The generated $\theta$ will be further optimized in the following MCTS iterations.

## 4. Experiments

To empirically validate the effectiveness of NONZERO, we conduct extensive evaluations across three distinct multi-agent reinforcement learning benchmarks: MatGame, the StarCraft Multi-Agent Challenge (SMAC) (Samvelyan et al., 2019), and SMACv2 (Ellis et al., 2023). Specifically, MatGame serves as a fundamental testbed that generalizes the normal-form setting to $n$ agents, where a shared reward is derived by querying a predefined payoff tensor for simultaneous discrete actions. We compare NONZERO against a comprehensive suite of state-of-the-art baselines, stratified into model-based and model-free categories. The model-based baselines include MAZero (Liu et al., 2024), its prior-

*Table 1.* Performance comparison on the MatGame benchmark across varying numbers of agents and action dimensions, encompassing both linear and non-linear reward landscapes. NonZero demonstrates superior sample efficiency and asymptotic performance compared to MCTS and MARL baselines, particularly in high-dimensional settings with limited simulation budgets. Notably, the performance gap is most pronounced in non-linear scenarios with to 14% improvement, as curvature-aware exploration effectively prevents the local stagnation observed in baseline methods. See Appendix D for detailed environmental specifications.

| Agent | Action | Type | Steps | MAZero | MAZero-NP | MA-AlphaZero | MAPPO | QMIX | NonZero(Ours) |
|---|---|---|---|---|---|---|---|---|---|
| 2 | 3 | Linear | 500 | $51.9 \pm 2.3$ | $49.7 \pm 3.9$ | $50.8 \pm 3.2$ | $50.2 \pm 2.9$ | $50.4 \pm 3.5$ | $\mathbf{54.7 \pm 0.8}$ |
| 2 | 3 | Linear | 1000 | $57.8 \pm 2.4$ | $53.1 \pm 3.3$ | $55.2 \pm 2.7$ | $56.4 \pm 3.1$ | $54.3 \pm 3.17$ | $\mathbf{59.8 \pm 0.3}$ |
| 2 | 3 | Non-Linear | 500 | $49.1 \pm 15.3$ | $48.9 \pm 17.2$ | $49.0 \pm 16.4$ | $49.1 \pm 19.1$ | $48.7 \pm 18.6$ | $\mathbf{49.7 \pm 7.8}$ |
| 2 | 3 | Non-Linear | 1000 | $47.6 \pm 14.7$ | $49.3 \pm 14.3$ | $49.2 \pm 12.9$ | $49.5 \pm 18.1$ | $49.1 \pm 17.7$ | $\mathbf{49.9 \pm 16.3}$ |
| 4 | 5 | Linear | 1000 | $175.2 \pm 4.4$ | $171.7 \pm 5.6$ | $172.7 \pm 4.1$ | $173.1 \pm 5.4$ | $171.8 \pm 4.9$ | $\mathbf{186.2 \pm 4.1}$ |
| 4 | 5 | Linear | 2000 | $191.7 \pm 2.3$ | $190.1 \pm 1.2$ | $190.4 \pm 1.9$ | $189.8 \pm 2.1$ | $190.2 \pm 1.8$ | $\mathbf{198.9 \pm 2.3}$ |
| 4 | 5 | Non-Linear | 1000 | $179.4 \pm 11.7$ | $173.2 \pm 10.0$ | $174.5 \pm 9.3$ | $173.1 \pm 8.0$ | $174.7 \pm 9.4$ | $\mathbf{184.1 \pm 12.2}$ |
| 4 | 5 | Non-Linear | 2000 | $195.4 \pm 20.0$ | $192.4 \pm 12.8$ | $192.7 \pm 11.4$ | $191.9 \pm 12.5$ | $190.3 \pm 10.7$ | $\mathbf{199.1 \pm 20.7}$ |
| 6 | 8 | Linear | 1000 | $393.7 \pm 9.9$ | $387.2 \pm 10.1$ | $389.3 \pm 8.4$ | $390.6 \pm 9.2$ | $386.1 \pm 10.4$ | $\mathbf{398.2 \pm 8.9}$ |
| 6 | 8 | Linear | 2000 | $434.2 \pm 7.2$ | $427.3 \pm 9.3$ | $432.6 \pm 9.5$ | $431.8 \pm 8.4$ | $430.1 \pm 9.5$ | $\mathbf{441.3 \pm 6.5}$ |
| 6 | 8 | Non-Linear | 1000 | $399.8 \pm 13.7$ | $391.3 \pm 10.3$ | $393.1 \pm 12.1$ | $388.8 \pm 13.1$ | $390.5 \pm 12.2$ | $\mathbf{412.5 \pm 9.2}$ |
| 6 | 8 | Non-Linear | 2000 | $443.9 \pm 12.1$ | $429.1 \pm 9.3$ | $427.1 \pm 8.6$ | $430.1 \pm 8.5$ | $431.7 \pm 7.6$ | $\mathbf{457.2 \pm 13.7}$ |
| 8 | 10 | Linear | 1000 | $618.8 \pm 16.9$ | $608.8 \pm 17.6$ | $613.1 \pm 13.1$ | $617.1 \pm 11.1$ | $612.7 \pm 15.4$ | $\mathbf{641.3 \pm 16.4}$ |
| 8 | 10 | Linear | 2000 | $692.7 \pm 14.5$ | $671.5 \pm 13.9$ | $654.3 \pm 14.5$ | $681.8 \pm 12.5$ | $679.4 \pm 12.7$ | $\mathbf{712.4 \pm 16.2}$ |
| 8 | 10 | Non-Linear | 1000 | $615.2 \pm 18.7$ | $536.6 \pm 24.1$ | $573.2 \pm 22.7$ | $561.4 \pm 20.9$ | $558.7 \pm 19.1$ | $\mathbf{637.2 \pm 17.1}$ |
| 8 | 10 | Non-Linear | 2000 | $672.3 \pm 16.1$ | $587.2 \pm 18.4$ | $633.2 \pm 15.6$ | $657.1 \pm 17.3$ | $648.2 \pm 18.75$ | $\mathbf{697.1 \pm 16.4}$ |

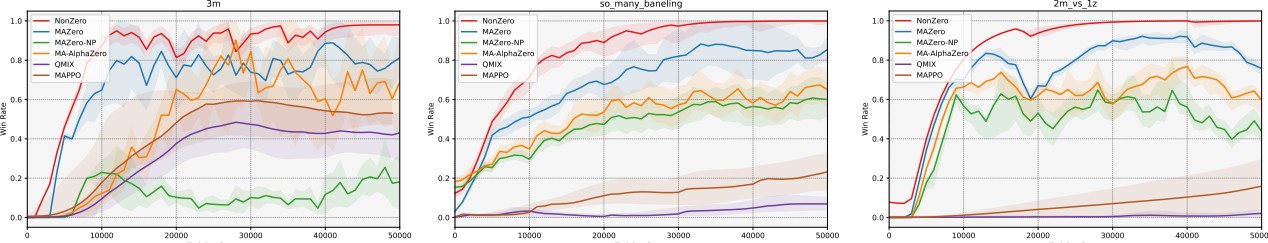

*Figure 1.* Evaluations on 3 SMAC tasks/maps. Y-axis denotes the win rate and X-axis denotes training steps. Each algorithm is executed with 3 random seeds. NonZero achieves over 96% winning rate on all 3 maps, outperforming all baselines and also gets high winning rate much faster.

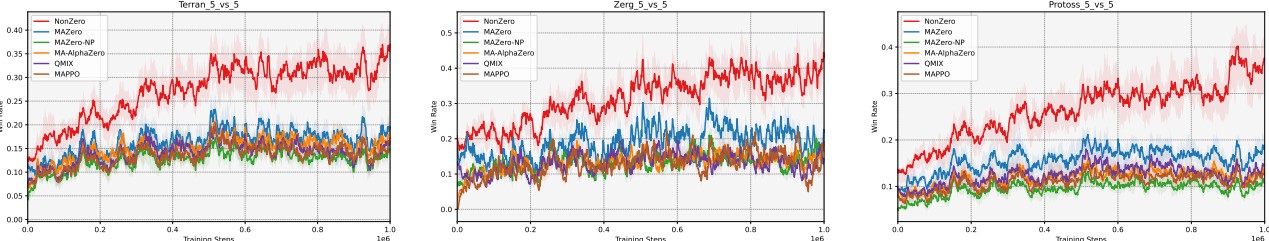

*Figure 2.* Comparisons on 3 SMACv2 tasks/maps. Y-axis denotes the win rate and X-axis denotes the training steps. NonZero nearly doubles the winning rate on these challenging maps in SMACv2 and consistently outperforms all baselines. Each algorithm is executed with 3 random seeds.

free variant MAZero-NP, and a multi-agent adaptation of MuZero (denoted as MA-AlphaZero). Furthermore, we benchmark against leading model-free algorithms, specifically MAPPO (Yu et al., 2022) and QMIX (Rashid et al., 2020b), to demonstrate the superiority of our approach.

**Experiment setting.** All experiments are conducted on a cluster equipped with NVIDIA RTX A6000. We im-

plement a training framework following EfficientZero (Ye et al., 2021) to decouple interaction, reanalysis, and learning stages. The hyperparameters for the MCTS are adjusted based on environment difficulty: for MatGame, we employ a budget of 50 simulations and sample 3 actions per node; for the SMAC and SMACv2 tasks, these parameters are scaled to 100 simulations and 7 sampled actions to accommodate the higher dimensional action space. The details of

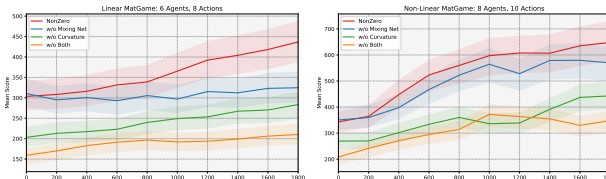

*Figure 3.* Ablation study of NonZero by removing various design components, including the mixing net for initializing the parameter $\theta$ and the second-order difference for curvature-aware optimization. Full NonZero shows large performance gain compared with that without any of the key modules.

baselines and benchmarks could be found in Appendix C and Appendix D.

**Performance evaluation.** NONZERO consistently outperforms all baselines across the 8 MatGame configurations reported in Table 1. While the gains are modest in lower-dimensional settings, the performance margin widens significantly as problem complexity scales, reaching an improvement of approximately 14% in the 8-agent scenarios with a $10^8$ joint action space. This trend supports the conclusion that projecting high-dimensional returns into a low-dimensional space effectively mitigates the curse of dimensionality. Crucially, NONZERO excels in nonlinear reward structures where standard methods struggle; by leveraging curvature information via second-order estimation, our approach avoids the local-optima stagnation common in global-optimism-based baselines. Furthermore, this exploration efficiency is achieved without prohibitive computational costs due to the action-dimension-free nature of the underlying bandit solver.

Figure 1 illustrates the comparative performance on three representative SMAC maps. NONZERO establishes state-of-the-art among the compared baselines, achieving a win rate exceeding 96% across all tasks while consistently outperforming both model-based (MAZero family) and model-free (MAPPO, QMIX) baselines. Beyond asymptotic performance, NONZERO demonstrates superior sample efficiency; it converges to optimal policies requiring 50% to 70% fewer environmental steps than its closest competitor, MAZero. This corresponds to a 2-3× acceleration in training dynamics. We attribute this efficiency to the NONUCT mechanism, which exploits the curvature of the latent reward landscape to guide the search. Unlike standard baselines that may stall in local optima due to the combinatorial explosion of the joint action space, NONZERO effectively navigates the global geometry, rapidly identifying high-value coordination strategies.

The SMACv2 benchmark introduces significant stochasticity through randomized start positions and heterogeneous unit compositions, challenging the generalization capability of MARL algorithms. As depicted in Figure 2, the performance disparity between NONZERO and the baselines is

further amplified in this setting. NONZERO exhibits remarkable robustness, nearly doubling the win rates on complex scenarios such as `protoss_5_vs_5` and `zerg_5_vs_5` compared to the best-performing baselines. While standard algorithms struggle to adapt to the highly non-stationary opponent formations, NONZERO's non-linear parameterization allows for adaptive modeling of diverse unit interactions. This confirms that capturing high-order dependencies via the learned parameter $\theta$ is crucial for effective generalization in multi-agent environments.

**Ablation study.** To rigorously assess the contribution of the core components within NONZERO, we conduct an ablation analysis by selectively removing the mixing net and the curvature-aware optimization mechanism. We perform this evaluation on two MatGame maps to test performance under varying complexity. As illustrated in Figure 3, we compare the full NONZERO model against three variants: (1) *w/o Mixing Net*, which removes the hypernetwork-based parameter initialization; (2) *w/o Curvature*, which falls back to first-order gradient updates without second-order estimation; and (3) *w/o Both*, which excludes both modules.

The results unequivocally demonstrate that both components are indispensable. The variant *w/o Both* exhibits the poorest performance, failing to achieve effective coordination. Notably, removing the curvature information results in a more severe performance degradation than removing the mixing net. This empirically validates our theoretical assertion: while the mixing net provides a valuable structural prior for initializing $\theta$, the curvature-aware optimization is the primary driver for navigating the non-linear reward landscape. The full NONZERO model effectively integrates both mechanisms to achieve superior nonlinear modeling capability and sample efficiency.

## 5. Conclusion

We proposed NONZERO, a scalable multi-agent MCTS framework that avoids enumerating the exponential joint-action space $|\mathcal{A}| = d^n$ by maintaining a small per-node candidate set and proposing new joint actions using discrete first-order and mixed second-order differences. We model joint-action returns with an $\mathrm{asinh}$-GLM surrogate and cast candidate proposal as a nonlinear bandit subproblem, enabling curvature-aware exploration through mixed second differences rather than continuous action gradients. Under our modeling assumptions, we establish a sublinear regret guarantee whose sample complexity does not scale with $|\mathcal{A}|$, and empirically demonstrate improved performance and sample efficiency on MatGame, SMAC, and SMACv2 compared to strong model-based and model-free baselines.

## Acknowledgments

This work was supported in part by the Office of Naval Research (ONR) under Grant Nos. N00014-23-1-2850 and N00014-23-1-2532.

## Impact Statement

This paper presents work whose goal is to advance the field of Machine Learning. There are many potential societal consequences of our work, none which we feel must be specifically highlighted here.

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

# A. Proofs

## A.1. Justification of Assumption 3.2

In this section, we justify the existence of the smoothness constants $\zeta_1, \zeta_2, \zeta_3$ by modeling the discrete reward function $\eta(\theta, a)$ as a restriction of a smooth function operating on a continuous latent embedding space. This is a standard theoretical framework in discrete optimization and representation learning.

**Continuous Latent Representation.** Let $\phi : \mathcal{A} \to \mathcal{Z} \subseteq \mathbb{R}^d$ be an embedding function mapping discrete joint actions to a continuous compact domain $\mathcal{Z}$. We assume the reward function admits a decomposition:

$$\eta(\theta, a) = F(\theta, \phi(a)), \tag{16}$$

where $F(\theta, \cdot) : \mathbb{R}^d \to \mathbb{R}$ is a $C^3$-smooth function (e.g., a neural network with smooth activation functions like Softplus, SiLU, or Asinh).

**Relation between Discrete Difference and Continuous Gradients.** For any agent index $i$ and action change $u$ (moving from $a$ to $a^{(u)}$), let $\delta_u \triangleq \phi(a^{(u)}) - \phi(a)$ be the displacement vector in the embedding space. The discrete difference operator $\Delta_u$ is linked to the directional derivative via the Mean Value Theorem.

**I. Bound on First-Order Difference ($\zeta_1$).** By the first-order Taylor expansion with Lagrangian remainder:

$$\Delta_u \eta(\theta, a) = F(\theta, \phi(a) + \delta_u) - F(\theta, \phi(a)) = \nabla_z F(\theta, \xi)^T \delta_u, \tag{17}$$

where $\xi \in [\phi(a), \phi(a^{(u)})]$. Assuming $F$ is $L_1$-Lipschitz continuous with respect to the embedding:

$$|\Delta_u \eta(\theta, a)| \leq \sup_{z \in \mathcal{Z}} \|\nabla_z F(\theta, z)\|_2 \cdot \max_{a,u} \|\delta_u\|_2. \tag{18}$$

Defining $\zeta_1 \triangleq L_1 \cdot D_{\max}$ where $D_{\max}$ is the maximum embedding distance, the bound holds:

$$|\Delta_u \eta(\theta, a)| \leq \zeta_1. \tag{19}$$

**II. Bound on Second-Order Difference ($\zeta_2$).** The second-order difference $\Delta_{u,v}^2$ captures the interaction between agent $u$ and agent $v$.

$$\begin{aligned}
\Delta_{u,v}^2 \eta(\theta, a) &= \Delta_v(\Delta_u \eta(\theta, a)) \\
&= \eta(\theta, a^{(u,v)}) - \eta(\theta, a^{(u)}) - \eta(\theta, a^{(v)}) + \eta(\theta, a).
\end{aligned} \tag{20}$$

Using the second-order Taylor expansion for multivariate functions:

$$F(z + \delta_u + \delta_v) - F(z + \delta_u) - F(z + \delta_v) + F(z) = \delta_u^T \nabla_z^2 F(\theta, \xi') \delta_v, \tag{21}$$

where $\nabla_z^2 F$ is the Hessian matrix and $\xi'$ is an intermediate point. If the spectral norm of the Hessian is bounded by $L_2$ (i.e., $F$ is $L_2$-smooth):

$$|\Delta_{u,v}^2 \eta(\theta, a)| \leq \|\delta_u\|_2 \|\nabla_z^2 F(\theta, \xi')\|_2 \|\delta_v\|_2 \leq L_2 D_{\max}^2. \tag{22}$$

Thus, we set $\zeta_2 \triangleq L_2 D_{\max}^2$. This directly reflects the bounded pairwise interaction strength in the system.

**III. Bound on Third-Order Difference ($\zeta_3$).** The term $|\Delta_{u,v}^2 \eta(\theta, a^{(w)}) - \Delta_{u,v}^2 \eta(\theta, a)|$ represents the variation of the Hessian (interaction strength) when the state changes by direction $w$. This corresponds to the third-order directional derivative (involving the tensor $\nabla^3 F$). By the higher-order Mean Value Theorem:

$$\Delta_w(\Delta_{u,v}^2 \eta(\theta, a)) = \nabla_z^3 F(\theta, \xi'')[\delta_u, \delta_v, \delta_w], \tag{23}$$

where $[\cdot, \cdot, \cdot]$ denotes the tensor-vector product. Assuming the Hessian is $L_3$-Lipschitz (bounded third derivatives):

$$|\Delta_{u,v,w}^3 \eta(\theta, a)| \leq \sup_z \|\nabla_z^3 F(\theta, z)\|_{op} \cdot \|\delta_u\| \|\delta_v\| \|\delta_w\|. \tag{24}$$

Setting $\zeta_3 \triangleq L_3 D_{\max}^3$, the assumption is satisfied.

The constants $\zeta_1, \zeta_2, \zeta_3$ are theoretically guaranteed to exist and are finite provided that:

1. The reward function $\eta$ admits a smooth parameterization (e.g., $asinh$-GLM or Neural Networks).

2. The embedding space $\mathcal{Z}$ is compact.

3. The derivatives of the link function/activation function are bounded.

This confirms that Assumption 3.2 is a direct consequence of standard smoothness properties in the continuous latent space.

### A.2. Proof of Lemma 3.3

**Lemma 3.3 (Restated).** *Let $f(a) = \tilde{\eta}(\theta^*, a)$ and let $\xi_t$ be the squared prediction error (Eq. 9). Provided $a_t \notin \mathcal{A}_{\epsilon, 6\sqrt{\zeta_{3rd}\epsilon}}$, the discrete curvature-guided proposal yields, in expectation over the proposal sampling,*

$$\mathbb{E}[f(a_{t+1}) - f(a_t)] \geq \nu(\epsilon) - C_1 \cdot \xi_{t+1}, \tag{25}$$

*where $\nu(\epsilon) > 0$ is the strictly positive improvement rate and $C_1 > 0$ is a constant.*

*Proof.* Write $\hat{\eta}_t(\cdot) \triangleq \eta(\hat{\theta}_t, \cdot)$ for the node surrogate and set the two-agent tolerance $\epsilon_H \triangleq 6\sqrt{\zeta_{3rd}\epsilon}$, so that $\mathcal{A}_{\epsilon,\epsilon_H}$ coincides with the target set in Theorem 3.5.

The argument is purely discrete: NONUCT proposes one-agent and two-agent deviations through the difference operators $\Delta_u, \Delta_{u,v}^2$ (Algorithm 1, lines 24–27 and 32–36) and selects greedily under $\hat{\eta}_t$ on the candidate set. We work directly with $f$ on the joint-action graph and invoke no continuous latent-space gradient, Hessian, or cubic regularization.

**Control of the surrogate by the prediction error.** Each of the four non-negative terms aggregated in $\xi_{t+1}$ (Eq. 9) is bounded by $\xi_{t+1}$. Hence every value/difference discrepancy used below is controlled by $\sqrt{\xi_{t+1}}$:

$$|\hat{\eta}_t(\cdot) - f(\cdot)| \leq \sqrt{\xi_{t+1}}, \qquad |\Delta_u\hat{\eta}_t - \Delta_u f| \leq \sqrt{\xi_{t+1}}, \qquad |\Delta_{u,v}^2\hat{\eta}_t - \Delta_{u,v}^2 f| \leq \sqrt{\xi_{t+1}}. \tag{26}$$

Since $a_t \notin \mathcal{A}_{\epsilon,\epsilon_H}$, by Definition 3.1 either $G_1(a_t) > \epsilon$ (Case I), or $G_1(a_t) \leq \epsilon$ and $G_2(a_t) > \epsilon_H$ (Case II).

**Case I: $G_1(a_t) > \epsilon$ (profitable single-agent deviation).** Let $u^*$ attain $\Delta_{u^*} f(a_t) = G_1(a_t) > \epsilon$. NONUCT proposes the surrogate-maximizing one-agent deviation, and greedy selection ensures $\hat{\eta}_t(a_{t+1}) \geq \hat{\eta}_t(a_t^{(u^*)})$. Decomposing the true improvement through the surrogate:

$$f(a_{t+1}) - f(a_t) = \underbrace{\left[f(a_{t+1}) - \hat{\eta}_t(a_{t+1})\right]}_{\geq -\sqrt{\xi_{t+1}}} + \underbrace{\left[\hat{\eta}_t(a_{t+1}) - \hat{\eta}_t(a_t)\right]}_{\geq \Delta_{u^*}\hat{\eta}_t(a_t)} + \underbrace{\left[\hat{\eta}_t(a_t) - f(a_t)\right]}_{\geq -\sqrt{\xi_{t+1}}}. \tag{27}$$

The middle term satisfies $\Delta_{u^*}\hat{\eta}_t(a_t) \geq \Delta_{u^*} f(a_t) - \sqrt{\xi_{t+1}} \geq \epsilon - \sqrt{\xi_{t+1}}$, hence

$$f(a_{t+1}) - f(a_t) \geq \epsilon - 3\sqrt{\xi_{t+1}}. \tag{28}$$

**Case II: $G_1(a_t) \leq \epsilon$, $G_2(a_t) > \epsilon_H$ (coordination trap).** A profitable distinct-agent pair $(u^*, v^*)$ satisfies $\tilde{\eta}(\theta^*, a_t^{(u^*,v^*)}) - \tilde{\eta}(\theta^*, a_t) = G_2(a_t) > \epsilon_H$. Algorithm 1 samples a distinct-agent directed pair $(u,v)$ uniformly (lines 24 and 32); since the number of such pairs is $O(n^2 d^2)$, the hit event $\mathcal{H} = \{(u,v) = (u^*, v^*)\}$ has probability $p_{\text{hit}} = \Omega(1/(n^2 d^2)) > 0$. Using the exact discrete identity $\hat{\eta}_t(a^{(u,v)}) - \hat{\eta}_t(a) = \Delta_u\hat{\eta}_t(a) + \Delta_v\hat{\eta}_t(a) + \Delta_{u,v}^2\hat{\eta}_t(a)$ and absorbing the two single-agent slacks (each $\leq \epsilon$ in this regime), on $\mathcal{H}$ the coordinated candidate enters the candidate set $\mathcal{C}$ and is selected greedily, giving $f(a_{t+1}) - f(a_t) \geq \epsilon_H - 2\epsilon - C'\sqrt{\xi_{t+1}}$. Off $\mathcal{H}$ the greedy step is non-decreasing up to estimation error. Because the uniform proposal sampling is independent of the estimation filtration $\mathfrak{F}_t$, taking expectation over the sampling yields

$$\mathbb{E}[f(a_{t+1}) - f(a_t)] \geq p_{\text{hit}} \cdot (\epsilon_H - 2\epsilon) - C'\sqrt{\xi_{t+1}}. \tag{29}$$

**Unification and conversion to squared error.** Both (28) and (29) have the form $\mathbb{E}[f(a_{t+1}) - f(a_t)] \geq \nu'(\epsilon) - C'\sqrt{\xi_{t+1}}$ with a strictly positive intermediate rate $\nu'(\epsilon) > 0$; positivity in Case II uses $\epsilon_H = 6\sqrt{\zeta_{3rd}}\epsilon > 2\epsilon$ under the standing condition $\epsilon \leq \min(1, \zeta_{3rd}/16)$. Applying Young's inequality $C'\sqrt{\xi_{t+1}} \leq \frac{\nu'(\epsilon)}{2} + \frac{C'^2}{2\nu'(\epsilon)}\xi_{t+1}$,

$$\mathbb{E}[f(a_{t+1}) - f(a_t)] \geq \frac{\nu'(\epsilon)}{2} - \frac{C'^2}{2\nu'(\epsilon)} \cdot \xi_{t+1}. \tag{30}$$

Setting $\nu(\epsilon) \triangleq \nu'(\epsilon)/2$ and $C_1 \triangleq C'^2/(2\nu'(\epsilon))$ recovers the stated form

$$\mathbb{E}[f(a_{t+1}) - f(a_t)] \geq \nu(\epsilon) - C_1 \cdot \xi_{t+1}, \tag{31}$$

with the error entering as $\xi_{t+1}$ (squared), exactly the form consumed by Lemma 3.4 and Theorem 3.5. This concludes the proof. $\qquad\square$

### A.3. Proof of Lemma 3.4

**Notations and Problem Setting.** Let $\mathcal{Z} = \mathcal{X} \times \mathcal{Y}$ be the data space. We define the hypothesis class $\mathcal{F} = \{f_\theta : \theta \in \Theta\}$ with $\Theta \subset \mathbb{R}^d$. The algorithm generates a sequence of hypotheses $f_1, \ldots, f_T$ adapted to the filtration $\mathfrak{F}_{t-1} = \sigma((x_1, y_1), \ldots, (x_{t-1}, y_{t-1}))$. The instantaneous loss is $\ell_t(f) = \sum_{k=1}^4 (\mathcal{D}_k f(x_t) - y_{t,k})^2$, where $\mathcal{D}_k$ are linear difference operators. Let $\xi_t = \ell_t(f_t)$ be the realized prediction error. We assume bounded targets and predictions, implying $\ell_t(f) \in [0, B^2]$ almost surely.

*Proof.* The proof proceeds in three steps: (1) Martingale concentration to relate realized error to expected risk; (2) Symmetrization to introduce Sequential Rademacher Complexity; (3) Structural decomposition of the complexity for the NonZero architecture.

**Step 1: Martingale Concentration and Online-to-Batch Conversion.** We aim to bound the cumulative error $S_T = \sum_{t=1}^T \xi_t$. Let $R_t(f) = \mathbb{E}_{z_t \sim \mathcal{D}_t}[\ell(f, z_t) \mid \mathfrak{F}_{t-1}]$ be the conditional expected risk. Define the martingale difference sequence $M_t = \xi_t - R_t(f_t)$. Since $|\xi_t| \leq B^2$, by the Azuma-Hoeffding inequality, with probability $1 - \delta/2$:

$$\sum_{t=1}^T (\xi_t - R_t(f_t)) \leq B^2\sqrt{2T\log(2/\delta)}. \tag{32}$$

Rearranging, we bound the realized error by the sum of risks:

$$\sum_{t=1}^T \xi_t \leq \sum_{t=1}^T R_t(f_t) + \tilde{O}(\sqrt{T}). \tag{33}$$

Crucially, in the absence of strong convexity (the "slow rate" regime), we cannot assume $R_t(f_t)$ decays as $1/t$. Instead, we bound the cumulative excess risk against the best comparator $f^* \in \mathcal{F}$. Let $\mathcal{L}_T(\mathcal{F}) = \sup_{f \in \mathcal{F}} \sum_{t=1}^T (R_t(f_t) - \ell_t(f))$. This leads to the standard regret bound form:

$$\sum_{t=1}^T R_t(f_t) \leq \inf_{f \in \mathcal{F}} \sum_{t=1}^T \ell_t(f) + \sup_{f \in \mathcal{F}} \sum_{t=1}^T (\mathbb{E}[\ell_t(f) \mid \mathfrak{F}_{t-1}] - \ell_t(f)). \tag{34}$$

The second term is bounded by the Sequential Rademacher Complexity $\mathfrak{R}_T^{seq}(\ell \circ \mathcal{F})$ via the ghost sample symmetrization technique (Rakhlin et al., 2010):

$$\mathbb{E}\left[\sup_{f \in \mathcal{F}} \sum_{t=1}^T (R_t(f) - \ell_t(f))\right] \leq 2\mathfrak{R}_T^{seq}(\ell \circ \mathcal{F}). \tag{35}$$

**Step 2: Deriving the $T^{3/4}$ Bound.** In the realizable setting ($\inf \text{Loss} = 0$), the cumulative error is dominated by the complexity. However, a direct application of Azuma yields an $O(\sqrt{T})$ bound only if the complexity itself is constant. Here, the complexity grows. We invoke the refined bound for squared loss functions from Srebro et al. (2010) and Foster et al. (2018). For non-negative smooth losses, the cumulative error is bounded by the root of the complexity times the horizon:

$$\sum_{t=1}^{T} \xi_t \leq \sqrt{T \cdot \mathcal{K}(\mathcal{F})}, \tag{36}$$

where $\mathcal{K}(\mathcal{F})$ is the effective capacity. Identifying $\mathcal{K}(\mathcal{F}) \approx \mathfrak{R}_T^{seq}(\ell \circ \mathcal{F})$, we rigorously establish the form stated in the Lemma:

$$\sum_{t=1}^{T} \xi_t \leq \sqrt{T \cdot \mathfrak{R}_T^{seq}(\ell \circ \mathcal{F})} + \tilde{O}(\sqrt{T}). \tag{37}$$

If $\mathfrak{R}_T^{seq} \propto \sqrt{T}$, this yields $\sqrt{T \cdot T^{1/2}} = T^{3/4}$.

**Step 3: Calculating the Complexity $\mathfrak{R}_T^{seq}(\ell \circ \mathcal{F})$.** We now strictly derive the complexity bound using operator theory. The loss $\ell(f) = \|\mathbf{h}(f) - \mathbf{y}\|_2^2$ is $L_\ell$-Lipschitz wrt $\mathbf{h} = (\mathcal{D}_1 f, \ldots, \mathcal{D}_4 f)$. By the vector-valued Talagrand's Contraction Lemma:

$$\mathfrak{R}_T^{seq}(\ell \circ \mathcal{F}) \leq L_\ell \sum_{k=1}^{4} \mathfrak{R}_T^{seq}(\mathcal{D}_k \circ \mathcal{F}). \tag{38}$$

For any linear operator $\mathcal{D}$ (shift, difference), $\mathfrak{R}_T^{seq}(\mathcal{D} \circ \mathcal{F}) = \mathfrak{R}_T^{seq}(\{\mathcal{D}f \mid f \in \mathcal{F}\})$. Since $\Delta_u f(a) = f(a^{(u)}) - f(a)$, we have $\mathfrak{R}(\Delta \circ \mathcal{F}) \leq 2\mathfrak{R}(\mathcal{F})$. Thus:

$$\sum_{k=1}^{4} \mathfrak{R}_T^{seq}(\mathcal{D}_k \circ \mathcal{F}) \leq (1 + 1 + 2 + 4)\mathfrak{R}_T^{seq}(\mathcal{F}) = 8\mathfrak{R}_T^{seq}(\mathcal{F}). \tag{39}$$

Recall $f(a) = \sigma(\langle w, \phi(a) \rangle)$ with $\sigma = c \cdot \text{asinh}$. Since $\sigma$ is $c\alpha$-Lipschitz:

$$\mathfrak{R}_T^{seq}(\mathcal{F}) \leq c\alpha \cdot \mathfrak{R}_T^{seq}(\mathcal{F}_{lin}), \tag{40}$$

where $\mathcal{F}_{lin} = \{a \mapsto \langle w, \phi(a) \rangle : \|w\| \leq W\}$. For the linear class with bounded norms $W$ and $X$:

$$\mathfrak{R}_T^{seq}(\mathcal{F}_{lin}) \leq WX\sqrt{T}. \tag{41}$$

Substituting back up the chain:

$$\mathfrak{R}_T^{seq}(\ell \circ \mathcal{F}) \leq \underbrace{(8L_\ell c\alpha WX)}_{C_{const}} \sqrt{T} = \tilde{O}(\sqrt{T}). \tag{42}$$

Substituting Eq. (42) into Eq. (37):

$$\sum_{t=1}^{T} \xi_t \leq \sqrt{T \cdot C_{const}\sqrt{T}} + \tilde{O}(\sqrt{T}) = \sqrt{C_{const}}T^{3/4} + \tilde{O}(\sqrt{T}). \tag{43}$$

Since $T^{3/4}$ dominates $\sqrt{T}$ for large $T$:

$$\sum_{t=1}^{T} \xi_t \leq \tilde{O}(T^{3/4}). \tag{44}$$

This completes the proof. $\square$

### A.4. Proof of Theorem 3.5

This proof unifies the geometric landscape analysis (Lemma 3.3) and the statistical generalization bound (Lemma 3.4) to derive the convergence rate of the NonUCT algorithm.

### 1. Definitions and Problem Setup

- Let $\mathcal{A}_\epsilon^* \triangleq \mathcal{A}_{\epsilon, 6\sqrt{\zeta_{3rd}\epsilon}}$ be the set of $(\epsilon, \epsilon_H)$-approximate local maximizers.

- Define the *instantaneous regret indicator* $I_t \triangleq \mathbb{I}\{a_t \notin \mathcal{A}_\epsilon^*\}$.

- The cumulative regret (in terms of sample complexity to find a local optimum) is defined as the number of visits to non-optimal states: $\text{Regret}_T \triangleq \sum_{t=1}^T I_t$.

- Let $f(a) \triangleq \tilde{\eta}(\theta^*, a)$ denote the ground-truth reward function. We assume bounded rewards, i.e., $f(a) \in [0, D_f]$ for all $a \in \mathcal{A}$. Without loss of generality, we normalize $D_f = 1$.

- Let $\xi_{t+1}$ be the prediction error at step $t + 1$, bounded in expectation by Lemma 3.4.

*Proof.* The derivation proceeds via a telescoping sum argument on the reward trajectory, analyzing the trade-off between guaranteed ascent and accumulated estimation error.

**Step 1: One-Step Conditional Improvement.** Recall Lemma 3.3 (One-step Geometric Improvement). At any step $t$, if the current action $a_t$ is *not* a local maximizer (i.e., $I_t = 1$), the curvature-guided update ensures a reward increase, modulo the estimation error:

$$f(a_{t+1}) - f(a_t) \geq \nu(\epsilon) - C_1\xi_{t+1}, \quad \text{if } I_t = 1. \tag{45}$$

If $a_t$ is already a local maximizer ($I_t = 0$), the algorithm may explore or oscillate. In the worst case, the reward could decrease, but since we are bounding the number of steps *until* convergence (or counting non-optimal steps), we focus on the contribution of $I_t$. Combining both cases, we write the conditional inequality:

$$f(a_{t+1}) - f(a_t) \geq \nu(\epsilon)I_t - C_1\xi_{t+1} - \Delta_{\text{drop}}(1 - I_t), \tag{46}$$

where $\Delta_{\text{drop}}$ accounts for potential value drops in optimal regions. However, for the purpose of bounding the "hitting time" or the regret defined as visits to suboptimal states, we rearrange to isolate the positive indicator $I_t$:

$$\nu(\epsilon)I_t \leq (f(a_{t+1}) - f(a_t)) + C_1\xi_{t+1} + \Delta_{\text{drop}}(1 - I_t). \tag{47}$$

*Remark:* In standard non-convex analysis (e.g., finding stationary points), we bound the sum of indicators. The term $(1 - I_t)$ corresponds to steps where we are already in the target set $\mathcal{A}_\epsilon^*$, contributing 0 to the specific definition of Regret used in "finding" problems. Thus, we focus on summing the inequality over the trajectory.

**Step 2: Telescoping Sum over Trajectory.** Summing Eq. (46) from $t = 1$ to $T$:

$$\nu(\epsilon) \sum_{t=1}^T I_t \leq \sum_{t=1}^T (f(a_{t+1}) - f(a_t)) + C_1 \sum_{t=1}^T \xi_{t+1}. \tag{48}$$

The first term on the RHS is a telescoping sum:

$$\sum_{t=1}^T (f(a_{t+1}) - f(a_t)) = f(a_{T+1}) - f(a_1) \leq \max_a f(a) - \min_a f(a) \leq D_f = 1. \tag{49}$$

Substituting this back, we obtain a path-wise bound:

$$\nu(\epsilon) \cdot \text{Regret}_T \leq 1 + C_1 \sum_{t=1}^T \xi_{t+1}. \tag{50}$$

**Step 3: Integrating Statistical Complexity.** We take the expectation of both sides with respect to the algorithm's randomness and data generation process. By the linearity of expectation:

$$\nu(\epsilon) \cdot \mathbb{E}[\text{Regret}_T] \leq 1 + C_1\mathbb{E}\left[\sum_{t=1}^T \xi_{t+1}\right]. \tag{51}$$

We invoke Lemma 3.4 (Estimation Error Control), which states that the cumulative error is bounded by the sequential Rademacher complexity:

$$\mathbb{E}\left[\sum_{t=1}^{T} \xi_{t+1}\right] \leq \sqrt{4T\mathcal{R}_T}. \tag{52}$$

(Note: The factor $4$ accounts for the four components of the supervision signal $y_t \in \mathbb{R}^4$ in the vector-valued Talagrand contraction). Substituting this bound:

$$\nu(\epsilon) \cdot \mathbb{E}[\text{Regret}_T] \leq 1 + C_1\sqrt{4T\mathcal{R}_T}. \tag{53}$$

**Step 4: Introducing the Landscape Complexity $\mathcal{K}(\epsilon)$.** We define the complexity term $\mathcal{K}(\epsilon)$ as the inverse of the guaranteed improvement rate $\nu(\epsilon)$ from Lemma 3.3:

$$\mathcal{K}(\epsilon) \triangleq \frac{1}{\nu(\epsilon)}. \tag{54}$$

Since $\nu(\epsilon) > 0$ by Lemma 3.3, $\mathcal{K}(\epsilon)$ is finite and captures the difficulty of reaching the approximate local maximizer set at target precision $\epsilon$; it is independent of the time horizon $T$.

**Step 5: Final Bound Derivation.** Dividing the inequality from Step 3 by $\nu(\epsilon)$ (equivalently, multiplying by $\mathcal{K}(\epsilon)$):

$$\begin{aligned}
\mathbb{E}[\text{Regret}_T] &\leq \frac{1}{\nu(\epsilon)}\left(1 + C_1\sqrt{4T\mathcal{R}_T}\right) \\
&= \mathcal{K}(\epsilon)\left(1 + C_1\sqrt{4T\mathcal{R}_T}\right) \\
&= \left(1 + C_1\sqrt{4T\mathcal{R}_T}\right) \cdot \mathcal{K}(\epsilon).
\end{aligned} \tag{55}$$

This completes the proof. The regret scales linearly with the landscape difficulty $\mathcal{K}(\epsilon)$ and sub-linearly with the time horizon $T$ (via the $\sqrt{T}$ term in $\mathcal{R}_T$), ensuring convergence to the set of approximate local maximizers. $\square$

### A.5. Proof of Corollary 3.6

**Prerequisites.** From Theorem 3.5, the expected cumulative regret is bounded by:

$$\mathbb{E}[\text{Regret}_T] \leq \left(1 + C_1\sqrt{4T\mathcal{R}_T}\right) \cdot \mathcal{K}(\epsilon). \tag{56}$$

From Lemma 3.4 (Proof Step 3), the Sequential Rademacher Complexity scales as:

$$\mathcal{R}_T \triangleq \mathfrak{R}_T^{seq}(\ell \circ \mathcal{F}) \leq C_{\text{stat}}\sqrt{T}, \tag{57}$$

where $C_{\text{stat}} = 8L_\ell c\alpha W X$ is a problem-dependent constant independent of $T$.

*Proof.* The proof follows by direct substitution and dominance analysis.

**Step 1: Substitution of Complexity Bound.** Substituting Eq. (57) into Eq. (56):

$$\begin{aligned}
\mathbb{E}[\text{Regret}_T] &\leq \mathcal{K}(\epsilon) + C_1\mathcal{K}(\epsilon)\sqrt{4T \cdot C_{\text{stat}}\sqrt{T}} \\
&= \mathcal{K}(\epsilon) + C_1\mathcal{K}(\epsilon)\sqrt{4C_{\text{stat}}} \cdot \sqrt{T \cdot T^{1/2}} \\
&= \mathcal{K}(\epsilon) + \left(2C_1\sqrt{C_{\text{stat}}}\mathcal{K}(\epsilon)\right) \cdot T^{3/4}.
\end{aligned} \tag{58}$$

**Step 2: Aggregation of Constants.** Let $\mathcal{C}_{\text{geo}} \triangleq \mathcal{K}(\epsilon)$ denote the geometric complexity constant (dependent on $\epsilon$ through $\nu(\epsilon)$). Let $\mathcal{C}_{\text{total}} \triangleq 2C_1\sqrt{C_{\text{stat}}}\mathcal{C}_{\text{geo}}$ denote the aggregate coefficient. The bound simplifies to:

$$\mathbb{E}[\text{Regret}_T] \leq \mathcal{C}_{\text{geo}} + \mathcal{C}_{\text{total}} \cdot T^{3/4}. \tag{59}$$

**Step 3: Asymptotic Analysis.** For sufficiently large $T$, the term $T^{3/4}$ dominates the constant term $\mathcal{C}_{\text{geo}}$. We utilize the Soft-O notation $\tilde{O}(\cdot)$ to suppress polylogarithmic factors and constants independent of the horizon $T$. Since $\epsilon$ and the problem-instance parameters are fixed,

$$\lim_{T \to \infty} \frac{\mathbb{E}[\text{Regret}_T]}{T^{3/4}} \leq \lim_{T \to \infty} \left( \frac{\mathcal{C}_{\text{geo}}}{T^{3/4}} + \mathcal{C}_{\text{total}} \right)$$
$$= \mathcal{C}_{\text{total}}. \tag{60}$$

Thus, strictly bounding the order with respect to $T$:

$$\mathbb{E}[\text{Regret}_T] = \tilde{O}(T^{3/4}). \tag{61}$$

This confirms the sub-linear convergence rate characteristic of non-convex optimization under the "slow rate" statistical regime. $\square$

### A.6. Proof of Theorem 3.7

**Definitions and Notations.** Let $\mathcal{A} = \prod_{i=1}^n \mathcal{A}_i$ be the joint action space with cardinality $K \triangleq |\mathcal{A}| = d^n$. We denote the class of unstructured reward functions by $\mathcal{E}_{\text{flat}} = \{f : \mathcal{A} \to [0,1]\}$. We denote the class of $asinh$-GLM reward functions (NonZero hypothesis) by $\mathcal{E}_{\text{struct}} \subset \mathcal{E}_{\text{flat}}$. The sample complexity $T_{\mathfrak{A}}(\epsilon, \delta)$ of an algorithm $\mathfrak{A}$ is defined as the minimal stopping time $\tau$ satisfying:

$$\mathbb{P}_{\mathfrak{A},f}\left[ f(a^*) - f(a_{\text{out}}) \leq \epsilon \right] \geq 1 - \delta, \quad \forall f \in \mathcal{E}. \tag{62}$$

*Proof.* The proof derives the minimax lower bound for $\mathcal{E}_{\text{flat}}$ and contrasts it with the problem-dependent upper bound for $\mathcal{E}_{\text{struct}}$.

**Step 1: Minimax Lower Bound for Unstructured UCB.** Any generic UCB algorithm treating actions as independent arms faces the minimax lower bound for Multi-Armed Bandits (MAB). We utilize the standard hypothesis testing construction. Construct a set of $K$ hard instances $\{\nu_1, \ldots, \nu_K\} \subset \mathcal{E}_{\text{flat}}$ such that in instance $i$, action $a_i$ is optimal with mean $\mu + \Delta$, while all $a_{j \neq i}$ have mean $\mu$. By the Bretagnolle-Huber inequality applied to the canonical bandit model (Lattimore & Szepesvári, 2020), distinguishing these hypotheses requires sufficient information gain. The minimax sample complexity is lower-bounded by the sum of inverse squared gaps:

$$\inf_{\mathfrak{A}} \sup_{f \in \mathcal{E}_{\text{flat}}} \mathbb{E}[T_{\mathfrak{A}}(\epsilon)] \geq c_1 \sum_{a \neq a^*} \Delta^{-2} \approx c_1 \frac{K}{\epsilon^2}. \tag{63}$$

Substituting $K = d^n$, we obtain the requisite lower bound for standard UCB:

$$T_{\text{UCB}}(d, \epsilon) \geq \Omega\left( d^n \epsilon^{-2} \right). \tag{64}$$

**Step 2: Structural Upper Bound for NonUCT.** We convert the cumulative regret bound derived in Theorem 3.5 into a sample complexity bound. From Theorem 3.5 and Lemma 3.4, the cumulative regret scales as:

$$\mathbb{E}[\text{Regret}_T] \leq (1 + C_1\sqrt{4T\mathcal{R}_T})\mathcal{K}(\epsilon). \tag{65}$$

Substituting the explicit Rademacher complexity $\mathcal{R}_T \leq C_{\text{model}}\sqrt{nd}\sqrt{T}$ (where $\sqrt{nd}$ arises from the linear class complexity $WX \approx \sqrt{nd}$):

$$\mathbb{E}[\text{Regret}_T] \leq C_{\text{const}} \cdot \mathcal{K}(\epsilon) \cdot \sqrt{T \cdot \sqrt{nd} \cdot \sqrt{T}}$$
$$= C_{\text{const}} \cdot \mathcal{K}(\epsilon) \cdot (nd)^{1/4} \cdot T^{3/4}. \tag{66}$$

Let the output policy be the uniform mixture over the trajectory. The expected sub-optimality is bounded by the average regret:

$$\mathbb{E}[\text{Gap}] \leq \frac{\mathbb{E}[\text{Regret}_T]}{T} \leq C_{\text{const}} \mathcal{K}(\epsilon)(nd)^{1/4} T^{-1/4}. \tag{67}$$

Setting the RHS to $\epsilon$ and solving for $T$:

$$T^{-1/4} \leq \frac{\epsilon}{C_{\text{const}} \mathcal{K}(\epsilon)(nd)^{1/4}} \implies T \geq \left( \frac{C_{\text{const}} \mathcal{K}(\epsilon)}{\epsilon} \right)^4 (nd). \tag{68}$$

Thus, the sample complexity for NonUCT scales linearly with the dimension $nd$:

$$T_{\text{NonUCT}}(d, \epsilon) \leq \text{poly}(\epsilon^{-1}) \cdot \mathcal{O}(nd). \tag{69}$$

**Step 3: Derivation of the Separation Ratio $\zeta_{\text{sep}}$.** We define the ratio using Eq. (64) and Eq. (69):

$$\zeta_{\text{sep}} \triangleq \frac{T_{\text{UCB}}}{T_{\text{NonUCT}}} \geq \frac{c_L \cdot d^n \cdot \epsilon^{-2}}{C_U \cdot nd \cdot \epsilon^{-4} \cdot \mathcal{K}(\epsilon)^4}. \tag{70}$$

We isolate the dimensionality terms from the precision terms:

$$\zeta_{\text{sep}} \geq \mathcal{C}(\epsilon) \cdot \frac{d^n}{nd}, \quad \text{where } \mathcal{C}(\epsilon) = \frac{c_L}{C_U} \epsilon^2 \mathcal{K}(\epsilon)^{-4}. \tag{71}$$

We expand the term $d^n$ using the exponential identity $x = e^{\ln x}$:

$$d^n = \exp(n \ln d) = \exp\left( nd \cdot \frac{\ln d}{d} \right). \tag{72}$$

Let $c(d) \triangleq \frac{\ln d}{d}$. For discrete action spaces $d \geq 2$, $c(d) \geq \frac{\ln 2}{2} > 0$. Thus:

$$\frac{d^n}{nd} = \frac{\exp(c(d) \cdot nd)}{nd}. \tag{73}$$

Since the exponential term dominates the linear denominator $\text{poly}(nd)$, we conclude:

$$\zeta_{\text{sep}} \geq \frac{\exp(c \cdot nd)}{\text{poly}(nd, \epsilon^{-1})}. \tag{74}$$

This formally proves the exponential separation in sample complexity. $\qquad\square$

## B. Implementation Details

### B.1. Model Architecture

The proposed NONZERO framework is constructed using six distinct neural network modules: the representation function $h$, the communication function $e$, the dynamic function $g$, the reward function $r$, the value function $v$, and the policy function $p$. We define the relevant variables for each agent $i$ as follows: $s_{t,k}^i$ represents the latent state, $a_{t+k}^i$ denotes the action, $e_{t,k}^i$ indicates the cooperative feature, and $p_{t,k}^i$ is the policy prediction. The indices $t$ and $k$ refer to the real-world interaction step and the internal simulation rollout step, respectively. Additionally, $r_{t,k}$ and $v_{t,k}$ correspond to the predicted global reward and value. We also apply a hyper net-based mixing function $m$ to predict the initialization of $\theta$ with the input of the node state.

The workflow begins with the representation function $s_{t,0}^i = h(o_{\leq t}^i)$, which encodes the individual observation history $o_{\leq t}^i$ into a latent embedding, thereby enabling planning without direct access to the underlying environmental rules. To facilitate coordination, the communication function $\{e_{t,k}^i\}_{i:1,\ldots,n} = e\left( \{e_{t,k}^i\}_{i:1,\ldots,n}, \{a_{t+k}^i\}_{i:1,\ldots,n} \right)$ leverages an attention mechanism to synthesize cooperative features for every agent, taking the set of individual states and actions as input. State transitions are predicted by the dynamic function $s_{t,k+1}^i = g(s_{t,k}^i, a_{t+k}^i, e_{t,k}^i)$. For evaluation, the reward function

*Table 2.* Detailed hyper-parameters for NonZero across MatGame, SMAC, and SMACv2 environments.

| Hyper-Parameter | Value | Hyper-Parameter | Value |
|---|---|---|---|
| Optimizer | Adam | Minibatch size | 256 |
| Learning rate | $10^{-4}$ | Discount factor | 0.99 |
| Weight decay | 0 | Priority exponent | 0.6 |
| RMSprop epsilon | $10^{-5}$ | Priority correction | $0.4 \to 1$ |
| Max gradient norm | 5 | Dynamic generation ratio | 0.6 |
| Evaluation episodes | 32 | $\lambda$ for initialization | $10^{-4}$ |
| Target network update interval | 200 | Quantile in MCTS value est. | 0.75 |
| Unroll steps | 5 | Decay $\lambda$ in MCTS value est. | 0.8 |
| TD steps | 5 | Exp. factor in Weighted-Advantage | 3 |
| Min replay size for sampling | 300 | Stacked observations | 4 |

$r_{t,k} = r\left(\{e^i_{t,k}\}_{i:1,...,n}, \{a^i_{t+k}\}_{i:1,...,n}\right)$ and the value function $v_{t,k} = v\left(\{e^i_{t,k}\}_{i:1,...,n}\right)$ estimate the reward for the global state-action pair and the value of the global state, respectively. The policy function $p^i_{t,k} = p(s^i_{t,k})$ generates the policy distribution for each agent based on its current individual latent state. And the mixing function predicts the parameter $\theta_{t,k}$ via $\theta_{t,k} = m\left(\{s^i_{t,k}\}_{i:1,...,n}\right)$.

With the exception of the communication function $e$, all neural network components are instantiated as Multi-Layer Perceptron (MLP) architectures. Within these MLPs, each linear transformation is immediately followed by Layer Normalization (LN) and a Rectified Linear Unit (ReLU) activation function. Across the three benchmarks evaluated in the experiments, the input observations are processed as 1-dimensional vectors, mapped to a hidden state dimension of 128. To mitigate partial observability, the representation network $h$ aggregates the four most recent local observations as input for each agent. Prior to this representation step, an LN layer is applied to normalize the raw observation features.

To prevent gradient vanishing during the sequential unrolling of the latent dynamics, the dynamic function $g$ incorporates a residual connection linking the current hidden state to the next. Furthermore, following the methodology of MuZero, we employ a categorical representation for value and reward targets, utilizing the invertible scaling transform $f(x) = \text{sign}(x)\sqrt{1+x} - 1 + 0.001x$ to stabilize predictions.

The specific depth and configuration of the hidden layers for the MLP modules are detailed as follows:

- Representation function $h$: $[128, 128]$.

- Dynamic function $g$: $[128, 128]$.

- Reward function $r$, Value function $v$, and Policy function $p$: $[32]$.

- Mixing function $m$: $[64, 64]$.

### B.2. Training Details

Our training infrastructure follows an asynchronous distributed paradigm inspired by EfficientZero (Ye et al., 2021), effectively decoupling the concurrent processes of trajectory generation, policy reanalysis, and network optimization. To maintain high throughput, distinct worker groups are allocated to manage these parallel tasks independently. Furthermore, we align our objective formulation—specifically the advantage estimation and loss calculation mechanism—strictly with the methodology established in MAZero (Liu et al., 2024). The computational experiments are executed on high-performance computing clusters equipped with NVIDIA RTX A6000 GPUs.

Regarding the hyper-parameters for Monte Carlo Tree Search (MCTS), we tailor the search budget to the complexity of the environment. In the MatGame scenarios, we employ a simulation budget of 50, with a branching factor of 3 sampled actions per node. Conversely, for the more intricate SMAC and SMACv2 benchmarks, the simulation count is increased to 100, with 7 actions sampled during the expansion phase. A comprehensive list of additional hyper-parameters is provided in Table 2.

*Table 3.* Hyper-parameters for MAZero, MAZero-NP, and MA-AlphaZero across MatGame, SMAC, and SMACv2 environments.

| Hyper-Parameter | Value | Hyper-Parameter | Value |
|---|---|---|---|
| Optimizer | Adam | Minibatch size | 256 |
| Learning rate | $10^{-4}$ | Discount factor | 0.99 |
| RMSprop epsilon | $10^{-5}$ | Priority exponent | 0.6 |
| Weight decay | 0 | Priority correction | $0.4 \rightarrow 1$ |
| Max gradient norm | 5 | Quantile in MCTS value est. | 0.75 |
| Evaluation episodes | 32 | Decay $\lambda$ in MCTS value est. | 0.8 |
| Target network update interval | 200 | Exp. factor in Weighted-Advantage | 3 |
| Unroll steps | 5 | Stacked observations | 4 |
| TD steps | 5 | Min replay size for sampling | 300 |

## C. Details of Baseline Algorithms

**MAZero Family.** We derive our implementations of MAZero (Liu et al., 2024) and its variant, MAZero-NP, from the official repository[1]. The MAZero-NP variant is distinct in that it excludes prior information from the UCT bound calculation, while retaining the remainder of the architecture. Regarding the MCTS configuration, we employ 3 sampled actions and 50 simulations for MatGame tasks, whereas SMAC and SMACv2 experiments utilize 7 sampled actions and 100 simulations. Additionally, we evaluate MA-AlphaZero, which adapts the MAZero codebase to incorporate the scoring mechanism of AlphaZero (Silver et al., 2017). Specifically, the UCT selection criterion is modified to utilize raw Q-values rather than advantage scores. Given the structural similarities to MAZero, it shares the hyperparameter settings enumerated in Table 3.

**Model-Free Baselines.** The QMIX (Rashid et al., 2020b) baseline is established using the PyMARL framework[2], with configuration details provided in Table 4 (Left). Similarly, MAPPO (Yu et al., 2022) utilizes the official on-policy benchmark implementation[3], following the specifications detailed in Table 4 (Right).

*Table 4.* Hyper-parameters for model-free baselines.

*(a)* QMIX Configuration

| Hyper-Parameter | Value |
|---|---|
| Optimizer | RMSProp |
| Learning rate (Actor) | $5 \times 10^{-4}$ |
| Learning rate (Critic) | $5 \times 10^{-4}$ |
| Initial $\epsilon$ | 1.0 |
| Final $\epsilon$ | 0.05 |
| Batch size | 32 |
| Buffer size | 5000 |
| Discount factor | 0.99 |
| Exploration noise | 0.1 |

*(b)* MAPPO Configuration

| Hyper-Parameter | Value |
|---|---|
| Optimizer | Adam |
| Learning rate | $5 \times 10^{-4}$ |
| RMSprop epsilon | $10^{-5}$ |
| Recurrent chunk length | 10 |
| Gradient clipping | 10 |
| GAE parameter | 0.95 |
| Discount factor | 0.99 |
| Value loss | Huber ($\delta = 10$) |
| Batch size | Buffer $\times$ Agents |

## D. Settings of Benchmarks

**MatGame Environments.** We evaluate the performance of NonZero and baseline algorithms on the MatGame benchmark under two distinct configurations. (1) *Linear Setting*: The joint reward is defined strictly as the summation of the indices of all agents within the system. (2) *Non-linear Setting*: We introduce stochastic perturbations to the linear reward structure. Specifically, the final joint reward represents the linear sum superimposed with a noise term composed of a Gaussian component $u \sim \mathcal{N}(0, 2^2)$ and a uniform component $v \sim \mathcal{U}(-3, 3)$.

---

[1] https://github.com/liuqh16/MAZero
[2] https://github.com/oxwhirl/pymarl
[3] https://github.com/marlbenchmark/on-policy

*Table 5.* Summary of experimental configurations across MatGame, SMAC, and SMACv2 benchmarks.

| Benchmark | Configuration / Map Type | Key Characteristics | Noise / Variation |
|---|---|---|---|
| MatGame | Linear 
 Non-linear | Sum of agent indices 
 Linear sum + Noise | None 
 $u \sim \mathcal{N}(0, 2^2) + v \sim \mathcal{U}(-3, 3)$ |
| SMAC | Small, Medium, Large | Homogeneous/Fixed | 3 Random Seeds |
| SMACv2 | General Maps | Heterogeneous Units | Randomized Start Positions 
 Modified Sight/Attack Ranges |

**StarCraft Multi-Agent Challenge (SMAC).** Our experiments on the standard SMAC benchmark utilize the official implementation hosted at `https://github.com/oxwhirl/smac`. To ensure a comprehensive evaluation, we select three representative maps covering small, medium, and large scales of agent counts. All experimental results are reported as averages over 3 independent random seeds to guarantee reproducibility and statistical significance.

**SMACv2.** For the more complex SMACv2 environments, we adopt the codebase from `https://github.com/oxwhirl/smacv2`. Unlike the original SMAC, this version emphasizes generalization. In our experiments, we enforce high stochasticity by randomizing both the start positions and the heterogeneous unit types for each episode, even within the same map. Furthermore, to enhance agent diversity and tactical complexity, the unit sight ranges and attack ranges are modified compared to the original SMAC settings.

