# OpenReview forum: "NonZero: Interaction-Guided Exploration for Multi-Agent Monte Carlo Tree Search"
_ICML.cc/2026/Conference — ICML 2026 spotlight_

### Official Review · Reviewer_NR8A · 2026-03-01

**Soundness:** 3
**Presentation:** 3
**Significance:** 4
**Originality:** 2
**Overall Recommendation:** 5
**Confidence:** 4

**Summary:**

The paper introduces NONZERO, a novel multi-agent Monte Carlo Tree Search (MCTS) framework designed to bypass the exponential branching factor ($|A|=d^n$) that typically bottlenecks cooperative multi-agent planning. Instead of relying on uninformed action sampling or enforcing restrictive structural assumptions (like strict additivity) on the value function, NONZERO drives tree expansion through a low-dimensional, nonlinear return surrogate.
The core contribution is the NONUCT proposal rule, which frames candidate expansion as a nonlinear bandit problem. NONUCT evaluates single-agent deviations for individual gains and utilizes a mixed second-order difference measure to detect non-additive interactions between two agents. This allows the algorithm to discover coordinated, multi-agent improvements and escape coordination traps where no single agent can unilaterally improve the return. The paper proves that NONUCT achieves a sublinear local-regret guarantee for reaching graph-local optima, with a sample complexity that avoids scaling with the full joint-action space. Empirically, the framework demonstrates superior sample efficiency and final performance against established baselines across several benchmarks.

**Compliance With Llm Reviewing Policy:**

Affirmed.

**Final Justification:**

The rebuttal successfully addressed the gap between the paper’s theory and its practical implementation. The work offers a mathematically sound approach to a difficult problem in MARL. I believe this paper makes a good contribution to the filed of RL and statistical decision making. I thus rate the paper at 'Accept (5)'.

**Key Questions For Authors:**

**Q1:** The asinh link function is introduced as a key structural design for satisfying the smoothness requirements. Have you tried other smooth continuous activations?

**Q2:** The regret bound relies on Assumption 3.2. In MARL environments with abrupt reward discontinuities, it feels like the constants bounding $\zeta_1, \zeta_2, \zeta 3$​ would explode, right? How does NONUCT behave (theoretically or empirically) when these smoothness assumptions are not met?


**Q3:** This is a minor concern, but the theoretical analysis requires the reward function to be C3-smooth over the latent embedding. Appendix A.1 assumes the Hessian is L3​-Lipschitz. However, in Appendix B.1, it is mentioned that the MLP modules use ReLU activation functions. ReLU cannot physically possess a Lipschitz continuous Hessian, right?

**Limitations:**

yes

**Strengths And Weaknesses:**

**Soundness**

Strengths: The theoretical result is sound. By targeting graph-local optimality rather than NP-hard global optimization in the huge combinatorial space, the paper presents a mathematically sound approach to bound cumulative regret. The paper also has some quite sound design in the theoretical aspects. The choice of the specific link function is good. It does not seem like any arbitrary choice: it is chosen because its controlled polynomial derivative decay satisfies the strict discrete smoothness assumptions (Assumption 3.2) required for the cubic regularization analysis.



**Presentation**

Strength: The presentation is good. The mathematical definitions and results are presented without ambiguity. The paper is also well-structured in how it maps a recognized MARL bottleneck (exponential branching) to a well-defined theoretical subproblem.


**Significance**

Strengths: The paper advances how MCTS can be applied to MARL. Prior SOTA relied heavily on static value factorization (e.g. VDN, QMIX), which enforces quite strict constraints (such as additivity or monotonicity) on the joint value function. NONZERO proves that you can capture complex, non-additive interactions dynamically during the MCTS expansion phase without needing exhaustive enumeration. To me, the significance lies mostly in its practical algorithmic utility for scaling MCTS. This probably doesn’t feel like a massive theoretical breakthrough, but still has good significance.

Weakness: The graph-local optimality guarantee (Theorem 3.5) feels a bit weak form of optimality in complex multi-agent settings: it only guarantees converging to a state where no single agent or pair of agents can improve.


**Originality**

Strengths: Treating tree expansion as an active optimization problem over a learned surrogate, and specifically using discrete mixed second-order differences to detect non-additive interactions, is original w.r.t. prior heuristic sampling methods.

Weakness: The technical novelty is weak. For example, the statistical and theoretical approach is heavily borrowed from existing continuous optimization literature rather than representing a foundational theoretical discovery in reinforcement learning. The nonlinear bandit formulation and the resulting regret bound seem standard derivations for slow-rate regimes.

---

> ### Author Rebuttal · Authors · 2026-03-29
>
> Thanks for your constructive suggestions. The clarification is provided below.
>
> Due to character limit, we just use number to denote the concern to spare more space for the clarification.
>
>
> **W1**
>
> We clarify why graph-local optimality is a well-motivated and strong guarantee in our setting:
>
> **(1) Local optima are near-global under our model class.** As noted in Section 3.2, the invexity properties of the asinh-GLM landscape ensure that approximate local maxima are effectively global or optimism-equivalent. Thus, convergence to an $(\\epsilon\_1, \\epsilon\_2)$-local maximizer is not merely a local statement---it provides a meaningful global quality proxy under our surrogate class.
>
> **(2) Action-dimension-free efficiency.** Theorem 3.5 achieves $\\tilde{O}(T^{3/4})$ regret *independent of* $|\\mathcal{A}|=d^n$, and Theorem 3.7 shows an exponential separation: standard UCB requires $\\exp(c \\cdot nd)$ samples while NonUCT needs only $\\mathrm{poly}(nd, \\epsilon^{-1})$. Extending to $k$-agent deviations would require $O(n^k d^k)$ evaluations per step, negating this tractability. The choice $k{=}2$ is the strongest guarantee preserving polynomial complexity.
>
> **W2**
>
> We acknowledge that our analysis draws on tools from nonlinear optimization, and we thank the reviewer for the opportunity to clarify where the technical novelty lies. Our contribution is not in the proof machinery per se, but in the problem formulation and the structural properties it yields for our novel multi-agent MCTS algorithm.
>
> We agree the $\\tilde{O}(T^{3/4})$ rate is characteristic of slow-rate regimes. However, the key property of Theorem 3.5 is that it is action-dimension-free: the bound does not scale with $|\\mathcal{A}|=d^n$. Standard nonlinear bandit bounds depend on the action-space size, because exploration must cover the full space. Our bound avoids this precisely because exploration operates over $O(nd)$ structured local deviations instead of $d^n$ joint actions. This action-dimension-free guarantee is, to our knowledge, the first of its kind for multi-agent MCTS under nonlinear rewards.
>
> **Q1**
>
> Yes. We evaluated several alternatives during development. Sigmoid-family links suffer from saturation: in high- or low-score regions, the discrete differences $\\Delta\_u \\eta$ and $\\Delta^2\_{u,v} \\eta$ collapse to near-zero, which deprives the curvature-guided proposal of useful signal. The asinh link occupies a favorable middle ground---it is unbounded and strictly monotone, yet infinitely differentiable with polynomially decaying derivatives like sigmoid, which enables both well-behaved smoothness constants and informative curvature signals across the entire score range.
>
> Here is an experiment that demonstrates the validity.
>
> | Map | Step | w/ Sigmoid | w/ asinh |
> |-|-|-|-|
> |2m\_vs\_1z|1e4|0.13±0.06|0.81±0.11|
> |2m\_vs\_1z|3e4|0.21±0.07|0.97±0.14|
> |2m\_vs\_1z|5e4|0.19±0.04|0.99±0.07|
>
> **Q2**
>
> We need to state that  $\\zeta\_1, \\zeta\_2, \\zeta\_3$ constrain the asinh-GLM surrogate $\\eta(\\theta, \\cdot)$, not the raw environment reward. So the constants will not explode. As shown in Appendix A.1, $\\zeta\_k$ depends on the $k$-th order derivative bound of the surrogate model and the maximum embedding distance $D\_{\\max}$. These are jointly determined by the smoothness of the asinh link, the compactness of the parameter space $\\Theta$, and the bounded norm of the n-hot action representation---all architectural properties. They remain finite regardless of the true reward's regularity.
>
> Our performance gain in experiments already covers the reviewer's concern. MatGame rewards are discrete payoff tensors---each joint action maps to an independent scalar with no continuity between neighbors, which is strictly more extreme than abrupt discontinuities.
>
> **Q3**
>
> The apparent contradiction arises from two distinct model components that we did not sufficiently distinguish in the text. The $C^3$-smoothness assumption and ReLU apply to different modules. The theoretical analysis (Appendix A.1) requires smoothness of the *node-wise asinh-GLM surrogate* $\\eta(\\theta, a) = c \\cdot \\mathrm{asinh}(\\alpha \\langle w(\\theta), \\psi(a) \\rangle)$. For any fixed $\\theta$, this is a composition of a linear function and the asinh link, which is $C^\\infty$---so the $C^3$ requirement is satisfied by construction. The ReLU activations mentioned in Appendix B.1 belong to the MuZero [1] planning modules (representation $h$, dynamics $g$, reward $r$, value $v$, policy $p$), which provide supervision targets (Eq. 6) but are not the object of the smoothness assumption. We acknowledge that the paper does not make this boundary explicit enough, and we will clarify that in the revised manuscript.
>
> Thanks again for your detailed and helpful comments.
>
> [1] Schrittwieser, Julian, et al. "Mastering Atari, Go, chess and shogi by planning with a learned model." Nature 588.7839 (2020): 604-609.

---

> > ### Author Rebuttal · Reviewer_NR8A · 2026-04-03
> >
> > Thank you to the authors for the comprehensive rebuttal. The clarification regarding Q3 is good; distinguishing between the smooth asinh-GLM surrogate used for the theoretical expansion rule and the ReLU-based MuZero modules used for supervision resolves my concerns regarding the Lipschitz constants and the Hessian analysis.
> >
> > Furthermore, the response to Q1 regarding the "Sigmoid vs. asinh" link function, supported by the new experimental results, effectively demonstrates the practical importance of the chosen architecture. I also appreciate the clarification on Weakness#1.
> >
> > These clarifications have resolved my main concerns regarding the technical novelty and the gap between the theoretical assumptions and the implementation. I decide to raise my score given the rebuttal.

---

> > > ### Author Response · Authors · 2026-04-08
> > >
> > > Thank you for the detailed comments and positive recommendations. We will further polish up our work.

---

### Official Review · Reviewer_T2w8 · 2026-03-12

**Soundness:** 3
**Presentation:** 4
**Significance:** 3
**Originality:** 3
**Overall Recommendation:** 5
**Confidence:** 2

**Summary:**

This paper proposes an approach for scaling MCTS under cooperative multi-agent settings. Rather than enumerating all joint actions, the authors maintains a small candidate set at each tree node and expands it by evaluating structured one-agent and two-agent deviations from current candidates. The candidate proposal step is formalized as a nonlinear bandit problem and an $\tilde{O}(T^{3/4})$ local regret guarantee is developed. Overall, a critical problem investigated by this manuscript is the combinatorial explosion of the joint-action space in multi-agent MCTS.

**Compliance With Llm Reviewing Policy:**

Affirmed.

**Final Justification:**

My concerns have been adequately addressed. I recommnd accept. This paper makes a meaningful contribution to multi-agent MCTS.

**Key Questions For Authors:**

What happens when the true reward function is poorly approximated by the asinh-GLM class?

MALinZero seems to be the closest prior work and is prominently discussed. Why was it excluded from experiments?

**Limitations:**

Yes.

**Strengths And Weaknesses:**

Strengths:
The regret bound in Theorem 3.5 scales with the statistical complexity of the surrogate class (polynomial in $nd$) rather than the combinatorial size $d^n$, and the efficiency separation in Theorem 3.7 formally quantifies the exponential gap against unstructured exploration. Simulation shows improvements on all three benchmarks. The ablation study clearly demonstrates that curvature information is more critical than the mixing network initialization.


Weaknesses:
The asinh-GLM model is a single-index model, the nonlinearity is applied to a scalar linear projection of the action features. This may be limited for $n$ agents with $d$ actions each. It is better to discuss the  functions outside the model class.
The connection between the bandit-theoretic analysis in Section 3.2 (which assumes a specific online learning protocol with greedy action selection) and the actual MCTS algorithm (Algorithm 1, which interleaves selection, expansion, and backup across a tree) is under-specified. How does the regret guarantee transfer from the single-node bandit setting to the full tree search?

---

> ### Author Rebuttal · Authors · 2026-03-29
>
> Appreciate your positive recommendation and we address the raised questions and weaknesses point-by-point.
>
> **W1(a): Expressiveness of the asinh-GLM and functions outside the model class**
>
> **Q1**
>
> We chose asinh over alternatives specifically for its analytical properties. We experimented with sigmoid-based links, but sigmoid's saturating gradients cause the discrete differences to vanish in high- or low-score regions, making curvature-based exploration ineffective. And it requires extra parameters to be tuned. The asinh function avoids this: it is strictly increasing, unbounded, and its derivatives decay polynomially rather than exponentially, which is precisely what enables the smoothness conditions in Assumption 3.2. Here is an experiment that compares the performance of NonZero with different link functions and it is demonstrated that asinh-GLM is more suitable.
>
> | Map | Step | w/ Sigmoid | w/ asinh |
> |-|-|-|-|
> |2m\_vs\_1z|1e4|0.13±0.06|0.81±0.11|
> |2m\_vs\_1z|3e4|0.21±0.07|0.97±0.14|
> |2m\_vs\_1z|5e4|0.19±0.04|0.99±0.07|
>
> The surrogate's role is to optimize the policy, not to precisely reconstruct the reward function. NonZero uses $\\eta(\\hat{\\theta}, \\cdot)$ to rank candidate deviations for proposing new joint actions into $\\mathcal{C}(s)$. For this purpose, we need the surrogate to assign higher scores to joint actions under better policies and lower scores to those under worse policies---as long as the relative rankings are preserved. This is a much weaker requirement than precisely modeling the reward function. A single-index model can preserve this ranking locally even when the true reward has multi-agent interaction structures that lie outside the asinh-GLM class. Under misspecification, the consequence is a wider convergence neighborhood (controlled by the irreducible approximation error $\\epsilon\_{\\text{approx}}$), not a failure of the algorithm. We believe this explains the significant performance improvement we observe, even for complex maps and heterogeneous team formations in SMAC evaluation. Thanks for the intriguing question.
>
> **W1(b): Connection between the bandit analysis and Algorithm 1**
>
> At each node $s$, the surrogate $\\hat{\\theta}\_s$ is initialized by the hypernetwork (line 23) during EXPAND as a warm start. The bandit's explore-exploit cycle occurs during BACKUP (lines 29--37): sampling deviation directions $u, v$ (line 32) and computing the reward differences (lines 35--36) corresponds to the bandit's arm selection and feedback; updating $\\hat{\\theta}\_s$ via gradient descent on $\\mathcal{L}\_{\\text{NonUCT}}$ (line 37) corresponds to the bandit's parameter update. The greedy selection in Theorem 3.5 corresponds to line 14 (SELECT), where the best candidate in $\\mathcal{C}(s)$ is chosen via the surrogate---this exploits the bandit's learned ranking. Each node maintains its own $\\hat{\\theta}\_s$ and $\\mathcal{C}(s)$, so the per-node bandit instances are self-contained.
>
> **W1(c): Regret transfer to the full tree**
>
> Our regret analysis follows the classical UCT paradigm [1] (the standard MCTS analysis), which derives per-node bandit regret bounds. The contribution of Theorem 3.5 is not a new tree-level analysis framework, but showing that the per-node bandit achieves action-dimension-free regret $\\tilde{O}(T^{3/4})$, replacing the $O(d^n)$-dependent bound of standard UCB.
>
> **Q2**
>
> Good suggestion. We add the comparison as follows to demonstrate the brought performance gain.
>
> | Agent | Action | Type | Steps | MALinZero | NonZero(Ours) |
> |-|-|-|-|-|-|
> |2|3|Linear|1000|59.9±0.2|59.8±0.3|
> |2|3|Non-Linear|1000|49.6±15.5|49.9±16.3|
> |4|5|Linear|2000|197.4±2.1|198.9±2.3 |
> |4|5|Non-Linear|2000|197.8±21.1|199.1±20.7|
> |6|8|Linear|2000|439.8±6.8|441.3±6.5|
> |6|8|Non-Linear|2000|451.1±12.8|457.2±13.7|
> |8|10|Linear|2000|705.2±15.7|712.4±16.2|
> |8|10|Non-Linear|2000|693.4±15.6|697.1±16.4|
>
> From the table above, we can see that our NonZero outperforms MALinZero in all scenarios on MatrixGame, except for achieving almost the same performance in one scenario with the simplest linear setting (where MALinZero is supposed to do well due to linearity).
>
> | Map | Step | MALinZero | NonZero(Ours) |
> |-|-|-|-|
> | 2m\_vs\_1z | 1e4 | 0.74±0.19 | 0.81±0.11 |
> | 2m\_vs\_1z | 3e4 | 0.94±0.08 | 0.97±0.14 |
> | 2m\_vs\_1z | 5e4 | 0.97±0.13 | 0.99±0.07 |
> | Protoss\_5\_vs\_5 | 1e5 | 0.14±0.05 | 0.16±0.04 |
> | Protoss\_5\_vs\_5 | 5e5 | 0.22±0.03 | 0.29±0.08 |
> | Protoss\_5\_vs\_5 | 1e6 | 0.36±0.05 | 0.38±0.05 |
>
> Here we report key points in the SMAC performance curve. Complete results will be added in the final version. It demonstrates that our NonZero outperforms MALinZero (as well as other compared algorithms in the manuscript) in multiple maps.
>
> Thanks again for your detailed comments.
>
> [1] Kocsis, Levente, and Csaba Szepesvári. "Bandit based Monte-Carlo planning." European Conference on Machine Learning. Springer, 2006.

---

> > ### Author Rebuttal · Reviewer_T2w8 · 2026-04-03
> >
> > I thank the authors for their detailed response and the additional experiments. The mapping between the bandit protocol and Algorithm 1 resolves my concern about the under-specified connection.
> >
> > I acknowledge that this paper makes a meaningful contribution to multi-agent MCTS by replacing combinatorial enumeration with dimension-efficient exploration. I would encourage the authors to tighten the misspecification discussion in the final version so the theoretical guarantees are transparently scoped.

---

> > > ### Author Response · Authors · 2026-04-08
> > >
> > > Thanks for your constructive comments and positive recommendations. We will revise the future version of our paper accordingly.

---

### Official Review · Reviewer_iQ7z · 2026-03-13

**Soundness:** 3
**Presentation:** 2
**Significance:** 3
**Originality:** 3
**Overall Recommendation:** 4
**Confidence:** 1

**Summary:**

This paper studies the combinatorial action-selection bottleneck in MCTS for cooperative multi-agent settings, where expanding the full joint-action space is infeasible. The authors propose NONZERO, a MuZero-style planning framework that maintains a small candidate set of joint actions per node and expands it using a learned nonlinear surrogate over joint-action returns, guided by first-order single-agent deviations and second-order mixed-difference interaction scores. The paper also introduces NONUCT, a proposal rule framed as a nonlinear bandit over local deviations, and provides a regret analysis toward approximate graph-local optima. Empirically, the method is evaluated on MatGame, SMAC, and SMACv2, where it is reported to outperform several model-based and model-free baselines under matched search budgets.

**Compliance With Llm Reviewing Policy:**

Affirmed.

**Final Justification:**

Thank the authors for the rebuttal. I have no more questions.

**Key Questions For Authors:**

The paper has a theory-to-algorithm mismatch. The actual procedure in Algorithm 1 is a discrete candidate-set search with local deviations, but the proofs rely on continuous latent-space gradients, Hessians, and cubic regularization.

**Limitations:**

yes

**Strengths And Weaknesses:**

**Strength:**
1. The paper tackles a real and important bottleneck in multi-agent planning.
2. The core intuition, using structured local deviations and explicit pairwise interaction scores to propose coordinated actions, is sensible and addresses a failure mode of purely single-agent local search.
3. The method is also reasonably well motivated from the planning perspective.
4. The paper includes multiple environments rather than a single toy domain.

**Weakness:**
1. The paper has a theory-to-algorithm mismatch. The actual procedure in Algorithm 1 is a discrete candidate-set search with local deviations, but the proofs rely on continuous latent-space gradients, Hessians, and cubic regularization.
2. The presentation should be polished further.

Overall, I am not an expert in this specific field and have no other questions. I think the paper has many theoretical analyses and a sound method design.

---

> ### Author Rebuttal · Authors · 2026-03-29
>
> Thank you very much for your positive recommendation of our work and your insightful comments. Following are our responses to your concerns.
>
> **W1 The paper has a theory-to-algorithm mismatch. The actual procedure in Algorithm 1 is a discrete candidate-set search with local deviations, but the proofs rely on continuous latent-space gradients, Hessians, and cubic regularization.**
>
> **Q1 The paper has a theory-to-algorithm mismatch. The actual procedure in Algorithm 1 is a discrete candidate-set search with local deviations, but the proofs rely on continuous latent-space gradients, Hessians, and cubic regularization.**
>
> We thank the reviewer and acknowledge that Lemma 3.3's proof using continuous tools (gradients, Hessians, cubic regularization) is unnecessary for the discrete algorithm. We provide a corrected purely discrete proof sketch that requires modifying only Lemma 3.3. It gives the same result in our analysis and closes this gap.
>
> NonZero has two layers where continuous/discrete tools apply differently:
>
> - Action selection (discrete): Algorithm 1 proposes candidates via discrete deviations $\\Delta_u \\eta$, $\\Delta^2_{u,v} \\eta$. No continuous gradient over actions is computed.
> - Parameter learning (continuous): $\\theta$ is updated via gradient descent on $\\mathcal{L}_{\\text{NONUCT}}$ (Eq. 7), where the Descent Lemma legitimately applies.
>
> The mismatch is that Lemma 3.3 concerns discrete improvement $f(a_{t+1}) - f(a_t)$, yet invokes continuous tools. The original proof is not without merit: Appendix A.1 establishes a valid bridge between discrete differences and continuous derivatives via the Mean Value Theorem, and the Descent Lemma naturally yields a quadratic error structure that avoids additional inequalities. However, this bridge was not made explicit within Lemma 3.3's proof, creating the appearance of a gap. Now we show below that a purely discrete proof recovers the same one-step bound $f(a_{t+1}) - f(a_t) \\geq \\nu(\\epsilon) - C_1 \\xi_{t+1}$, preserving compatibility with Lemma 3.4 and Theorem 3.5.
>
> Let $f(a) = \\tilde{\\eta}(\\theta^*, a)$ and $\\xi_t$ be the squared prediction error (Eq. 9). Suppose $a_t \\notin \\mathcal{A}_{\\epsilon, \\epsilon_H}.$
>
> Case I: ($G_1(a_t) > \\epsilon$): A profitable single-agent deviation $u^\*$ exists with $\\Delta_{u^*} f(a_t) > \\epsilon$. We decompose the real improvement via the surrogate:
>
> \\[f(a_{t+1}) - f(a_t) = \\underbrace{[f(a_{t+1}) - \\eta(\\hat{\\theta}, a_{t+1})]}\_{\\geq -\\sqrt{\\xi_{t+1}}} + \\underbrace{[\\eta(\\hat{\\theta}, a_{t+1}) - \\eta(\\hat{\\theta}, a_t)]}\_{\\geq \\Delta_{u^*}\\eta(\\hat{\\theta}, a_t)} + \\underbrace{[\\eta(\\hat{\\theta}, a_t) - f(a_t)]}\_{\\geq -\\sqrt{\\xi_{t+1}}}.\\]
>
> The middle term satisfies $\\Delta_{u^\*}\\eta(\\hat{\\theta}, a_t) \\geq \\Delta_{u^\*} f(a_t) - \\sqrt{\\xi_{t+1}} \\geq \\epsilon - \\sqrt{\\xi_{t+1}}$, giving:
>
> $$
> f(a_{t+1}) - f(a_t) \\geq \\epsilon - 3\\sqrt{\\xi_{t+1}}.
> $$
>
> Case II: ($G_1 \\leq \\epsilon$, $G_2(a_t) > \\epsilon_H$): A profitable two-agent pair $(u^\*,v^\*)$ exists. Algorithm 1 randomly samples $(u,v)$, hitting $(u^\*,v^\*)$ with probability $p_{\\text{hit}} = \\Omega(1/(n^2 d^2))$. By the same surrogate-decomposition, in expectation:
>
> $$
> \\mathbb{E}[f(a_{t+1}) - f(a_t)] \\geq p_{\\text{hit}} \\cdot (\\epsilon_H - 2\\epsilon) - C'\\sqrt{\\xi_{t+1}}.
> $$
>
> Converting to squared error via Young's inequality. Combining both cases yields the intermediate bound $\\mathbb{E}[f(a_{t+1}) - f(a_t)] \\geq \\nu'(\\epsilon) - C'\\sqrt{\\xi_{t+1}}$ with $\\nu'(\\epsilon) = \\Theta(\\epsilon \\cdot p_{\\text{hit}})$. Applying Young's inequality $C'\\sqrt{\\xi_{t+1}} \\leq \\frac{\\nu'(\\epsilon)}{2} + \\frac{C'^2}{2\\nu'(\\epsilon)}\\xi_{t+1}$:
>
> $$
> \\mathbb{E}[f(a_{t+1}) - f(a_t)] \\geq \\frac{\\nu'(\\epsilon)}{2} - \\frac{C'^2}{2\\nu'(\\epsilon)} \\cdot \\xi_{t+1}.
> $$
>
> Setting $\\nu(\\epsilon) = \\nu'(\\epsilon)/2$ and $C_1 = C'^2/(2\\nu'(\\epsilon))$, we recover the original statement:
>
> $$
> \\boxed{f(a_{t+1}) - f(a_t) \\geq \\nu(\\epsilon) - C_1 \\cdot \\xi_{t+1}} \\quad \\square.
> $$
>
> The error is now $\\xi_{t+1}$ (squared), exactly matching the form required by Lemma 3.4 and Theorem 3.5. Since $p_{\\text{hit}}$ is polynomial in $nd$, the adjusted constants change $K(\\epsilon)$ by at most polynomial factors in $nd$, preserving the action-dimension-free property ($\\text{poly}(nd)$ vs. $d^n$).
>
>
>
>
> **W2 The presentation should be polished further.**
>
> We will polish our paper presentation for the following version, including providing more detailed explanation for the proof part.
>
> Thanks again for your helpful comments. We will try our best to improve the quality of our work.

---

> > ### Author Rebuttal · Reviewer_iQ7z · 2026-04-04
> >
> > Thank you for the rebuttal. I am not an expert in this area, and I have no more questions.

---

> > > ### Author Response · Authors · 2026-04-08
> > >
> > > Thanks for your helpful comments and positive recommendations. We will improve the future version of our paper.

---

### Decision · Program_Chairs · 2026-04-30

**Decision:**

Accept (spotlight)

**Comment:**

This paper introduces NONZERO, a novel multi-agent Monte Carlo Tree Search (MCTS) framework designed to bypass the exponential branching factor that typically bottlenecks cooperative multi-agent planning. Specifically, NONZERO drives tree expansion through a low-dimensional, nonlinear return surrogate. The core contribution is the NONUCT proposal rule, which frames candidate expansion as a nonlinear bandit problem. Theoretically, this paper proves that NONUCT achieves a sub-linear local-regret guarantee for reaching graph-local optima, with a sample complexity avoiding scaling with the full joint-action space. Empirically, the framework demonstrates superior sample efficiency and final performance against established baselines across several benchmarks.

My understanding is that this paper has significantly advanced how MCTS can be applied to MARL. This paper proves that you can capture complex, non-additive interactions dynamically during the MCTS expansion phase without exhaustive enumeration. I agree with Reviewer NR8A that the paper's significance lies mostly in its practical algorithmic utility for scaling MCTS. I think that the presentation of this paper is good, and to the best of my knowledge, the theoretical analyses are also sound.

All reviewers recommend accepting this paper. After reading the paper, the reviews, and the rebuttals, I agree with the reviewers and also recommend accepting this paper.